# Autoregulation of the LIM kinases by their PDZ domain

Gabriela Casanova-Sepúlveda[1], Joel A. Sexton [2], Benjamin E. Turk [2] & Titus J. Boggon [1,2] ✉

LIM domain kinases (LIMK) are important regulators of actin cytoskeletal remodeling. These protein kinases phosphorylate the actin depolymerizing factor cofilin to suppress filament severing, and are key nodes between Rho GTPase cascades and actin. The two mammalian LIMKs, LIMK1 and LIMK2, contain consecutive LIM domains and a PDZ domain upstream of the C-terminal kinase domain. The roles of the N-terminal regions are not fully understood, and the function of the PDZ domain remains elusive. Here, we determine the 2.0 Å crystal structure of the PDZ domain of LIMK2 and reveal features not previously observed in PDZ domains including a core-facing arginine residue located at the second position of the 'x-Φ-G-Φ' motif, and that the expected peptide binding cleft is shallow and poorly conserved. We find a distal extended surface to be highly conserved, and when LIMK1 was ectopically expressed in yeast we find targeted mutagenesis of this surface decreases growth, implying increased LIMK activity. PDZ domain LIMK1 mutants expressed in yeast are hyperphosphorylated and show elevated activity in vitro. This surface in both LIMK1 and LIMK2 is critical for autoregulation independent of activation loop phosphorylation. Overall, our study demonstrates the functional importance of the PDZ domain to autoregulation of LIMKs.

Cytoskeletal remodeling occurs in response to external stimuli and is required for essential processes such as cell invasion, proliferation, cytokinesis, adhesion, and differentiation[1–3]. Actin severing is necessary for a dynamic cytoskeleton and is regulated by the LIM (Lin11, Isl-1 & Mec-3) domain kinases (LIMK), which are key effectors of Rho GTPase pathways[4–7]. Multiple Rho-effector protein kinases, including the ROCK, PAK and MRCK groups phosphorylate and activate the LIMKs[8–10]. Importantly, the LIMKs (and the TESKs which are related in their catalytic domains[11]) appear unique in their ability to phosphorylate residue serine-3 of the actin depolymerizing factor, cofilin, which results in its inactivation[8,11–17]. This unique high-fidelity kinase-substrate recognition therefore provides an essential link between Rho GTPase activation and suppression of actin severing, placing the LIMKs as central nodes in the many cellular processes for which elongation of

actin filaments are required. Nonetheless, despite their importance, many details of LIMK function remain to be revealed, including the mechanism by which these proteins are autoregulated.

LIMKs are found in most animal species but are absent from fungi and plants, in humans their expression profiles differ, with LIMK1 showing higher expression in the brain, kidney, lung, stomach and testis, and LIMK2 with broader expression in both adult and embryonic tissue. LIMKs across species have a common architecture, with two N-terminal tandem-zinc finger LIM domains followed by a PDZ domain, a predicted unstructured region enriched in serine, proline and glycine residues, and a C-terminal kinase domain (Fig. 1a). Like many other kinases, activation of these multi-domain enzymes is associated with phosphorylation of the kinase activation loop at a conserved residue (Thr-508 of human LIMK1 and Thr-505 of human

[1]Department of Molecular Biophysics and Biochemistry, Yale University, New Haven, CT 06520, USA. [2]Department of Pharmacology, Yale University, New Haven, CT 06520, USA. ✉e-mail: titus.boggon@yale.edu

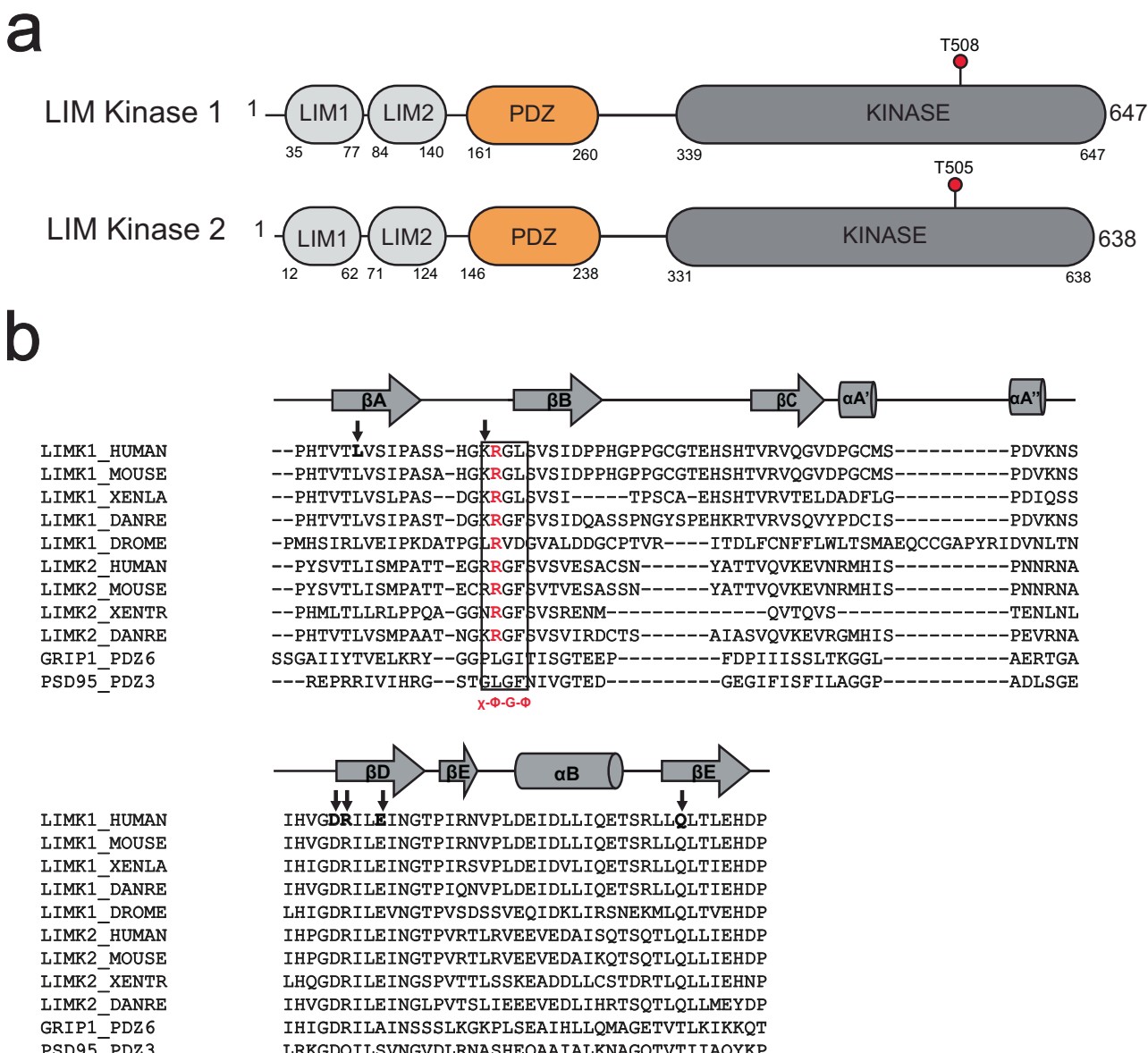

**Fig. 1 | LIMK domain architecture. a** LIM domain kinase family architecture showing human LIMK1 (UniProt ID: P53667), and human LIMK2 (UniProt ID: P53671). LIM1: first LIM domain, LIM2: second LIM domain, PDZ: PDZ domain, Kinase: kinase domain. Activation loop phosphorylation residues indicated, Thr508/Thr505 for LIMK1 and LIMK2, respectively. Residue numbers are shown. **b** Sequence alignment of PDZ domains. Alignment was created using PROMALS[74]. Uniprot ID for LIMK1_HUMAN, P53667; LIMK1_MOUSE, P53668; LIMK1_XENLA,

O42565; LIMK1_DANRE, B3DIV5; LIMK1_DROME, Q8IR79; LIMK2_HUMAN, P53671; LIMK2_MOUSE, O54785; LIMK2_XENTR, F7AFJ1; LIMK2_DANRE, Q6DG29; GRIP1_PDZ6, P97879; PSD95_PDZ3, P31016. GRIP1_PDZ6 is a Class I PDZ domain, PSD95_PDZ3 is a Class II PDZ domain. 'x-Φ-G-Φ' sequence is inside a black lined box. The conserved arginine residue equivalent to Arg163 in human LIMK1 is colored red. Conserved amino acid residues targeted in mutagenesis studies are in bold and under a black arrow.

LIMK2)[8–10,15]. LIMK activation loop phosphorylation is considered incompatible with its autoinhibited state, but the molecular basis for autoinhibition of the LIMKs remains unknown. Early studies suggested that the N-terminal domains play roles in regulation of catalytic activity. For example, truncated LIMK has elevated activity compared to full-length protein in vitro and in cultured cells, and the N-terminal non-catalytic region diminished catalytic activity of the isolated kinase domain in trans[10,18,19]. Similarly, mutation of the activation loop threonine to an unphosphorylatable residue results in suppression of activity, but phosphomimetic mutation increases activity[8,9]. Yet, despite these findings the molecular basis for this suppression of activity remains unclear.

The PDZ domain was named after its early identification in three proteins (postsynaptic density 95, PSD-95; discs large, Dlg; zonula occludens-1, ZO-1)[20–24], and more than 250 examples have been found in over 150 human proteins[25–27]. Generally, these non-catalytic domains are thought to mediate protein-protein interactions, typically by specific recognition of linear peptide motifs in the C-terminal tails of protein binding partners[28–33]. The LIMK PDZ domain is unusual however in that it does not interact tightly with carboxy-terminal peptides[27,34]. PDZ domains can also mediate protein interactions through alternate modes, including interactions of the canonical binding site with internal peptide motifs, or use of alternative binding surfaces[35–42]. Many of the differences between canonical and non-

canonical PDZ domains focus on a central binding site between two structural features of the domain, an α-helix and a β-strand (Supplementary Fig. 1a) and the abilities of non-canonical PDZ domains to bypass, modify or control these features (Supplementary Fig. 1b–e). There is, therefore, a possibility that the LIMK PDZ domain might similarly use alternative binding surfaces for intermolecular protein-protein interactions, however interactions with potential binding partners have not been identified[43–47]. Early studies suggested that the PDZ domain may impact LIMK autoregulation[18,19], but it is still unclear whether this occurs through canonical interactions with the peptide binding cleft or through other binding surfaces.

In this study we provide an in-depth analysis of the PDZ domain of the LIMK family. We determined the 2.0 Å crystal structure of the LIMK2 PDZ domain and found a canonical PDZ fold with an unusually shallow peptide binding cleft. We also found a highly conserved surface distal to the canonical peptide binding cleft, suggesting an unusual non-canonical mechanism of function for the LIMK PDZ. Targeted mutagenesis of the conserved surface, but not the canonical peptide binding site, resulted in elevated kinase activity in vitro and enhanced growth suppression when LIMK1 was ectopically expressed in yeast and elevated activation loop phosphorylation. We conclude that the LIMKs contain an unusual PDZ domain that plays a direct role in autoinhibition of kinase activity via a previously unidentified conserved surface. These findings shed light on the mechanism of regulation of LIMKs.

## Results

### LIMKs contain a divergent PDZ domain

To explore the role of the LIMK PDZ domain, we generated a sequence alignment of the PDZ domain from 421 LIMK orthologs across animal species with a set of canonical PDZ domains from other proteins. We found high conservation of the LIMK PDZ domain between human LIMK1 (residues 159-258) and human LIMK2 (residues 147-239), which are 47% identical and 81% similar. This high conservation is maintained across species, with the human LIMK1 PDZ being 36% identical to that of the *D. melanogaster* ortholog for example. There was lower sequence similarity to canonical PDZ domains, (21% identical to PSD95). Interestingly, one of the defining features of canonical PDZ domains was divergent in all LIMK orthologs; this motif is termed the 'G-L-G-F' motif (after a sequence in the PSD-95 protein) or more generally termed the 'x-Φ-G-Φ' motif, where x represents any, and Φ represents hydrophobic amino acid[24,48,49] (Fig. 1b, Supplementary Fig. 2). To investigate how the unique conserved features of the LIMK PDZ primary sequence related to its structure, we undertook X-ray crystallography analysis of the LIMK2 PDZ domain.

We expressed, purified, and crystallized the human LIMK2 PDZ domain (residues 145-236), which is monomeric in solution, and determined its structure to 2.0 Å resolution (Fig. 2a, Table 1). The crystal structure revealed a compact globular domain resembling a partially open barrel that is typical of the PDZ fold. We observed the expected canonical six β-strands and the canonical αB helix, but unusually, found that helix αA is replaced by two $3_{10}$ helices, which we term αA' (residues R187-H189) and αA" (residues P192-N194). In addition, we found a third $3_{10}$ helix in the βD-αB loop that we term the αB' helix (residues V212-T214) (secondary structure nomenclature as per[48]). We observed good electron density throughout the structure (Supplementary Fig. 3) and that crystal packing appeared to induce a conformational movement in the βB-βC loop resulting in two orientations visible in the asymmetric unit (Supplementary Fig. 4). Dali searches with the two orientations revealed that the LIMK2 PDZ domain was most similar to the PDZ domains of spinophilin (RMSDs of 3.0 Å and 3.5 Å over 88 and 88 Cαs for the two LIMK orientations; PDB ID: 3EGG[50]), syntenin-1 (2.5 Å/2.5 Å over 78/80 Cαs; PDB ID: 5G1E[51]), disks large homolog 4 (2.8 Å/2.5 Å over 87/85 Cαs; PDB ID: 5HEY[52]) and harmonin (3.0 Å/2.8 Å over 85/84 Cαs; PDB ID: 3KLR[53]). The structure

of LIMK2 PDZ domain thus corresponded overall to a canonical PDZ domain with unusual features.

For canonical PDZ domains, recognition of terminal carboxylate groups is 'conferred by a cradle of main chain amides'[48] contributed by the x-Φ-G-Φ motif, where x is any residue, and Φ is any hydrophobic residue. Unusually, the LIMKs do not follow this consensus sequence. Instead, they harbor KRGL and RRGL sequences in LIMK1 and LIMK2 respectively, replacing the first hydrophobic residue with a conserved arginine residue, Arg163 (Fig. 1b). Alignment over all human PDZ domains indicated that the LIMKs are the only PDZ domains harboring an arginine residue in the second position of the x-Φ-G-Φ motif. The residue at this position is normally oriented toward the hydrophobic core of the domain. Unusually for a charged residue, we found Arg163 in a similar orientation. To balance the charge of the guanidino group, Arg163 engages in extensive hydrogen bonding. It caps helix αB, hydrogen bonds to the carboxyl oxygens of residues Ala223, Ile224 and Gln226, and also hydrogen bonds to the carboxyl oxygen of Gln229 within the αB-βF loop (Fig. 2b, c). This arrangement seems to provide a rigid anchor for the C-terminus of the αB helix. A consequence of this inward-facing arginine residue is that it helps create a somewhat shallow binding grove between the βB strand and αB helix (Fig. 2d, Supplementary Fig. 5). Canonical PDZ domains utilize the βB-αB groove to bind partner peptides and coordinate terminal carboxylate groups through backbone amide interactions of the central Φ-G residues of the 'x-Φ-G-Φ' motif. The inward orientation of Arg163 to cap helix αB seems to be key for orientations of the βA-βB and αB-βF loops, and crystal packing does not seem to impact these orientations. In addition, an inward orientation of helix αB and placement of Arg163's Cβ atom to encroach on the expected carboxylate binding site provides a potential explanation for why the LIMK PDZ domains do not interact with carboxy-terminal peptides with measured affinities in a biological range[27,34].

### LIMK PDZ domains contain an extended conserved surface

Considering the unusual nature of the completely conserved Arg163, we wondered whether a more detailed conservation analysis could highlight the role of the LIMK PDZ domain. We therefore mapped conservation from our alignment of 421 LIMK sequences onto our crystal structure. Supporting our conjecture that the orientation and interactions of Arg163 may help preclude carboxy-terminal peptide interactions, we did not find complete conservation of the canonical βB-αB binding groove over all the LIMKs (Fig. 3a) or for individual conservation mapping of LIMK1 or LIMK2 (Supplementary Fig. 6). In contrast, we discovered almost complete conservation of an extended surface distal to the βB-αB binding groove comprising parts of strands βA, βF and βD (Fig. 3a). Based on calculated electrostatic potential, this surface is largely hydrophobic (Fig. 3b). This highly conserved nature of this βA-βF-βD surface suggested that it has a functional role across the LIMK family, either by structurally stabilizing the protein, or by participating in inter- or intra-molecular interactions. We therefore decided to study a potential autoregulatory role of this surface of LIMK.

### Mutation of conserved PDZ surface suppresses yeast growth

To evaluate LIMK autoregulation in living cells, we modified our previously reported system in which we reconstituted the mammalian LIMK1-cofilin pathway in budding yeast. The sole yeast cofilin ortholog (Cof1) is essential for viability, and expression of mammalian cofilin-1 can rescue the growth of a *cof1Δ* strain[54,55]. We have shown previously that expression of the LIMK1 catalytic domain suppresses the growth of yeast expressing human cofilin in a manner dependent on Ser3 phosphorylation[11]. We hypothesized that if full-length (FL) LIMK1 is autoinhibited by its N-terminal region, then it would cause a less severe growth phenotype when expressed in yeast in comparison to the catalytic domain alone. Furthermore, we assume that mutations in the

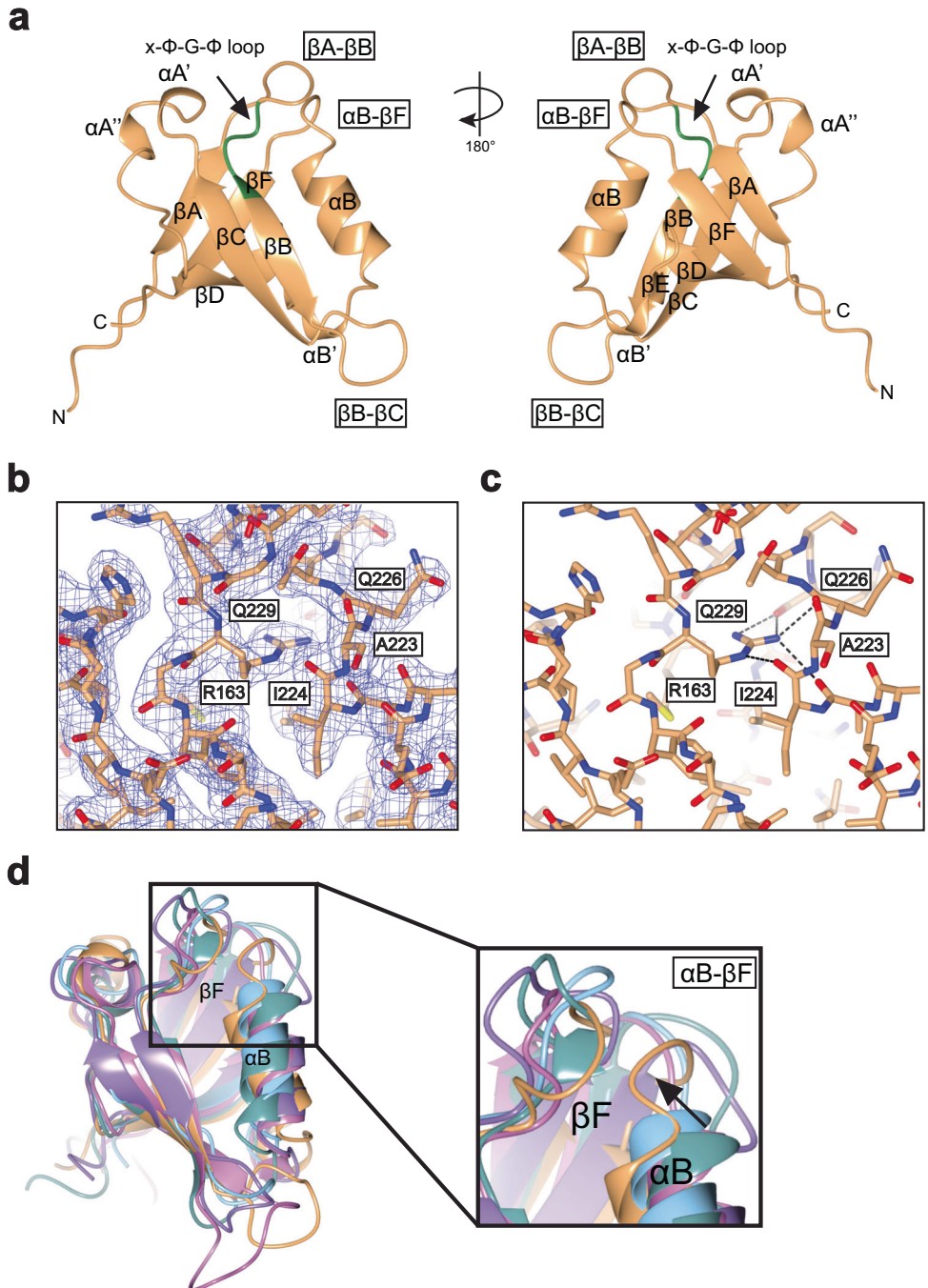

**Fig. 2 | Structure of LIMK2 PDZ domain. a** Human LIMK2 PDZ domain determined to 2.0 Å resolution shown in cartoon format. Secondary structure named. The 'x-Φ-G-Φ' loop, βA-βB loop, αB-βF loop and βB-βC loop are indicated and the x-Φ-G-Φ loop colored green. **b** Electron density map of Arg163. $2F_{obs}-F_{calc}$ electron density map contoured at 1σ (blue). $F_{obs}-F_{calc}$ electron density map contoured at +3σ (green) and −3σ (red). **c** Hydrogen bonds of Arg163. **d** Inward orientation of the αA-βF loop. Comparison of the αA-βF loop orientation of LIMK2 PDZ crystal structure (orange), and the most similar PDZ domains structures from Dali[75]; spinophilin, PDB ID: 3EGG[50] (pink), disks large homolog 4, PDB ID: 5HEY[52] (light blue), harmonin, PDB ID: 3K1R[53] (purple), and syntenin-1, PDB ID: 5G1E[51] (teal). Inward orientation of helix αB helps create a somewhat shallow binding grove between the βB strand and αB helix compared to these most similar PDZ domains. Images generated using CCP4mg[76].

PDZ domain that relieve autoinhibition will exacerbate growth suppression by FL LIMK1. We therefore used this *S. cerevisiae* system to assess the impact of mutations in the βA-βF-βD surface of the LIMK PDZ domain.

We transformed *cof1Δ* yeast with two plasmids, one constitutively expressing human cofilin-1, and the other expressing WT LIMK1 or various mutants thereof in a galactose-inducible manner. We then examined cell growth under conditions that either induce (galactose) or do not induce (glucose) LIMK expression. In contrast to induction of LIMK1 kinase domain expression, which resulted in complete growth suppression, expression of FL LIMK1 reduced, but did not completely eliminate growth. These observations suggest decreased cofilin phosphorylation by the presumably lower kinase activity of FL LIMK1 (Fig. 4a)[11]. We found no reduction in growth for cofilin-S3A expressing

**Table 1 | Data collection and refinement statistics**

| Data Collection | LIMK2 PDZ |
|---|---|
| PDB accession code | 8GI4 |
| Wavelength (Å) | 0.97918 |
| Resolution range (Å) | 80.42 – 2.06 (2.13–2.04) |
| Space group | $P 2_1$ |
| Cell dimensions a, b, c (Å) | 80.9 83.0 83.1 |
| α, β, γ (°) | 90, 96.6, 90 |
| Unique reflections | 67631 (6631) |
| Multiplicity | 20.2 (14.1) |
| Completeness (%) | 99.8 (98.3) |
| Mean $I/\sigma I$ | 23.9 (2.0) |
| Wilson B factor (Å²) | 45.8 |
| $R_{pim}$ | 4.5 (40.2) |
| CC½ | 99.5 (0.3) |
| CC* | 99.9 (0.68) |
| **Refinement** | |
| Resolution range (Å) | 80.42 – 2.06 (2.13 – 2.06) |
| Reflections used in refinement | 67544 (6630) |
| Reflections used for $R_{free}$ | 3197 (271) |
| % Reflections used for $R_{free}$ | 4.7 (4.1) |
| $R_{work}$ (%) | 21.0 (36.2) |
| $R_{free}$ (%) | 23.3 (36.6) |
| No. of non-hydrogen atoms | |
| Protein | 6091 |
| RMSD | |
| Bond lengths (Å) | 0.002 |
| Bond angles (°) | 0.45 |
| Ramachandran plot | |
| Favored, allowed, outliers (%) | 98.1, 2.0, 0.0 |
| Rotamer outliers (%) | 0 |
| MolProbity clashscore | 1.5 (100th percentile) |
| Average B factor (Å²) | 59.8 |

Statistics for the highest resolution shell are shown in parentheses. RMSD: root-mean-square deviation.

yeast upon induction of either kinase domain or FL LIMK1, confirming that growth suppression is dependent on cofilin Ser3 phosphorylation and not due to non-specific toxicity (Supplementary Fig. 7).

We then analyzed our crystal structure of the LIMK2 PDZ domain and assessed the conservation and solvent exposure of residues within the conserved the βA-βF-βD surface. Based on the high sequence similarity of LIMK1 and LIMK2 in the PDZ domain (Fig. 4b, S2) we introduced point mutations to alter the surface electrostatics or hydrophobicity (L165A, E225A, D221A, R222A, Q251A; human LIMK1 numbering) of the conserved βA-βF-βD patch. We then assessed the impact of these mutations on yeast growth. We found that all five point mutations increased LIMK1-dependent growth suppression, with E225A (equivalent to LIMK2 E206A) resulting in complete loss of growth, suggesting LIMK activity comparable to the uninhibited kinase domain (Fig. 4a). All mutant constructs expressed to the same level as FL LIMK1, suggesting appropriate folding (Supplementary Fig. 7).

**PDZ mutations increase LIMK catalytic activity**

To assess whether these alterations in yeast growth were indeed due to changes in LIMK catalytic activity, we examined the level of cofilin phosphorylation following LIMK induction by Phos-tag SDS-PAGE. We observed that PDZ domain mutations increased the proportion of phosphorylated cofilin in yeast (Fig. 4c, d). As this analysis suggested

increased kinase activity, we directly assessed the impact of LIMK mutations on phosphorylation of cofilin in vitro. We purified FL WT LIMK1 and the panel of PDZ domain mutants alongside a catalytic domain control from yeast. In kinase activity assays with these LIMK preparations, we found that K175D, a non-conserved mutant, showed no difference in cofilin phosphorylation compared to the WT FL LIMK1. In contrast, most mutants showed a significant increase in kinase activity, with E225A having the highest increase (Fig. 5). Solubility analysis for the PDZ domain alone suggests that D221A and R222A are destabilizing but that Q251A and E225A remain soluble, potentially indicating divergent mechanisms for changes in LIMK activity (Supplementary Fig. 8). Overall, we infer that the conserved βA-βF-βD surface of the PDZ domains of LIM domain kinases represents a surface that can impact LIMK kinase activity.

We finally assessed the role of the PDZ domain in the regulation of LIMK activation loop phosphorylation. The steps of regulation for these kinases are not resolved, and it is still unclear how autoregulation and activation loop phosphorylation coordinate to regulate activity. Therefore, we wondered if introduction of these point mutations could impact the phosphorylation of the LIMK activation loop. We found that in keeping with coordinated intramolecular interactions, activation loop phosphorylation was consistently elevated for point mutations that increased kinase activity. We observed higher activation loop phosphorylation in cell lysates from our yeast growth assays (Fig. 6) as well as in purified protein used for our kinase assays (Supplementary Fig. 9). To examine whether increased activation loop phosphorylation could account for elevated activity of PDZ domain mutants, we introduced point mutations into the activation loop that have previously been shown to mimic (T508EE; replacement of threonine-508 with two glutamic acids), phosphorylation-associated alterations in LIMK catalytic activity. In contrast, although we found that introduction of the phosphomimetic mutation T508EE into the activation loop of LIMK1 suppressed growth to a lesser extent than WT LIMK1 (presumably because it does not fully simulate phosphorylation), addition of E225A resulted in reduced viability, suggestive of increased catalytic activity (Fig. 7a, b). Similarly, the compound mutant LIMK1 (E225A/T508EE) displayed stronger phosphorylation of cofilin in vitro than the activation loop phosphomimetic mutation alone (Fig. 7c).

## Discussion

The LIM domain kinases are critical regulators of cytoskeletal dynamics in animal cells. They recognize and phosphorylate ADF/cofilin proteins by a non-canonical mechanism, and the near-monogamous kinase-substrate relationship makes the LIMKs fundamental for regulation of actin filament stabilization. The regulatory mechanisms by which the LIMKs are themselves controlled remain poorly described. In this study we provide insights into the molecular basis of LIMK autoregulation using a structure-directed approach focused on the LIMK PDZ domain. We determined the 2.0 Å crystal structure of the LIMK2 PDZ domain, which revealed key differences differentiating it from other PDZ domains. Our structure-based conservation mapping onto the LIMK2 PDZ domain revealed a previously unappreciated highly conserved surface patch and lead us to investigate the role of this region in autoregulation. We introduced point mutations in this surface based on our analysis of sequence conservation and surface electrostatics. Disruption of this site resulted in increased LIMK catalytic activity as demonstrated by in vivo yeast assay and in vitro kinase assay. Our work provides insights into the basis for LIMK regulation and highlights a conserved surface on the LIMK PDZ domain as a critical component of the regulatory mechanisms for the LIM domain kinases.

Our crystal structure reveals structural insights into the well-studied PDZ fold. Comparison of the LIMK2 PDZ domain to other human PDZ domains revealed three unusual features suggestive of

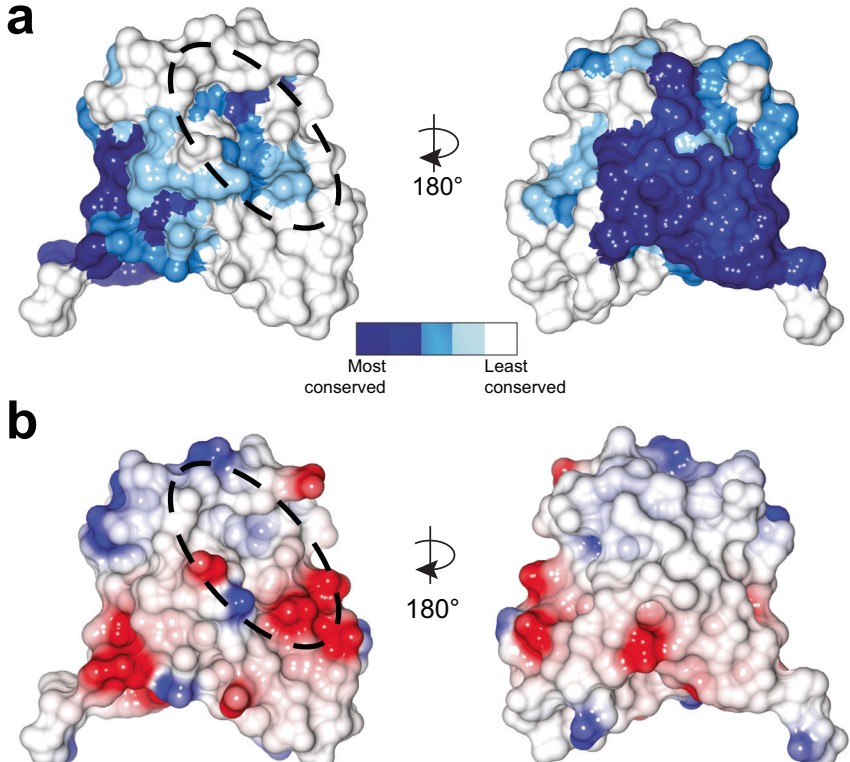

**Fig. 3 | Surface analysis of LIMK2 PDZ domain. a** Conservation of the LIMK2 PDZ domain. PDZ domain conservation mapped onto the structure of LIMK2 PDZ for 421 aligned LIMK sequences from mammals, birds, fish, and insects. Dashed oval indicates the canonical PDZ binding cleft. **b** Surface electrostatics of the LIMK2 PDZ domain calculated by CCP4mg[76]. Red indicates negatively charged surfaces, blue indicates positively charged surfaces, and white surfaces indicate neutrally charged surfaces. Dashed oval indicates the canonical PDZ binding cleft.

functional relevance. First, we observed that the canonical peptide binding cleft between the βB strand and αB helix is particularly shallow, and that the orientation of the αB-βF loop encroaches on the binding grove. While it is not necessarily unusual to observe a shallow cleft in PDZ domains (for example PDZ7 of GRIP[42]) this feature provides a rationale for why the LIM domain kinase PDZ domains have so far not been found to interact with C-terminal peptides with biological range affinities in PDZ interaction screening studies[27,34]. Second, we found that the second position of the 'x-Φ-G-Φ' motif was unique in the entire PDZ fold – a hydrophobic core-facing arginine residue (Arg163 in LIMK2, and Arg176 in LIMK1). The hydrogen-bonding interactions of this stringently conserved arginine caps the αB helix, coordinates the αB-βF loop, and seems to provide a rigid base for the C-terminus of the αB helix. Third, we found that the αA helix is replaced by two $3_{10}$ helices. This combination of unusual features for the LIMK PDZ domain make it difficult to place into the previously assigned PDZ classes (classes I, II, III or IV[37,56–59]) (Supplementary Fig. 1a). These features do, however, tempt conjecture that this domain could engage in bi-directional allostery. Previous studies (e.g. the interaction between Cdc42 and Par6[60,61]) have found that binding partner interactions, often with helix αA, can increase carboxylate peptide binding affinity and vice versa (Supplementary Fig. 1b−e). It is therefore interesting to speculate that the LIMK PDZ domain may be primed for carboxylate peptide binding, but require allosteric-induced conformational movements to reveal the high-affinity binding site. Further studies will be needed to probe this more fully.

Our structure also provides insight into the current status of macromolecular structure prediction. Comparison of our crystal structure with the NMR structure of LIMK2 PDZ domain (Riken Structural Genomics Proteomics Initiative; PDB ID: 2YUB) reveals that some of the unique features of the LIMK PDZ domain were not found

by NMR, including the $3_{10}$ helices, αA′ and αA″. Furthermore, the buried 'x-Φ-G-Φ' arginine, Arg-163, is surface exposed in the majority of the 20 NMR models (17/20). In contrast, AlphaFold (model AF-P53671-F1-model_v2.pdb[62]) predicts both of the $3_{10}$ helices, αA′ and αA″, and the buried Arg-163 (Supplementary Fig. 10). Our molecular replacement solution of the crystal structure was more accurate using the AlphaFold model than the NMR structure, (TFZ scores of 28 versus 7, respectively), and the final structure (chain A) displays RMSDs of 0.76 Å over 91 Cαs and 1.42 Å over 89 Cαs when compared to the AlphaFold and NMR (model 1) structures, respectively. These analyses suggest that AlphaFold can provide near-experimental accuracy for molecular models of folded domains even when unique structural features are present.

As is common among protein kinases, release from autoregulation is associated with LIMK activation loop phosphorylation[6,8–10], but the details of how the LIMKs are autoregulated remains unclear. Early studies suggested a 'head-tail' interaction between the N-terminal LIM and PDZ domains and the C-terminal kinase domain[10,18,19], and the activity of the catalytic domain alone is ~10-fold higher than the full-length protein[18,19]. Our study begins to provide some molecular level details on this regulation mechanism. Unexpectedly, we observed a lack of conserved residues in the canonical βB-αB cleft, but high conservation on the βA-βF-βD surface. The importance of this βA-βF-βD surface in LIMK regulation has previously not been established, and our introduction of point mutations result in increased catalytic activity of the full-length protein, consistent with disruption of an autoinhibited conformation. Our studies strongly imply that the βA-βF-βD surface, and particularly a conserved glutamic acid, E206 (LIMK2) / E225 (LIMK1), are critical for autoregulation, and that this regulation seems to be independent of activation loop phosphorylation. Importantly, we found that surface mutations outside of this

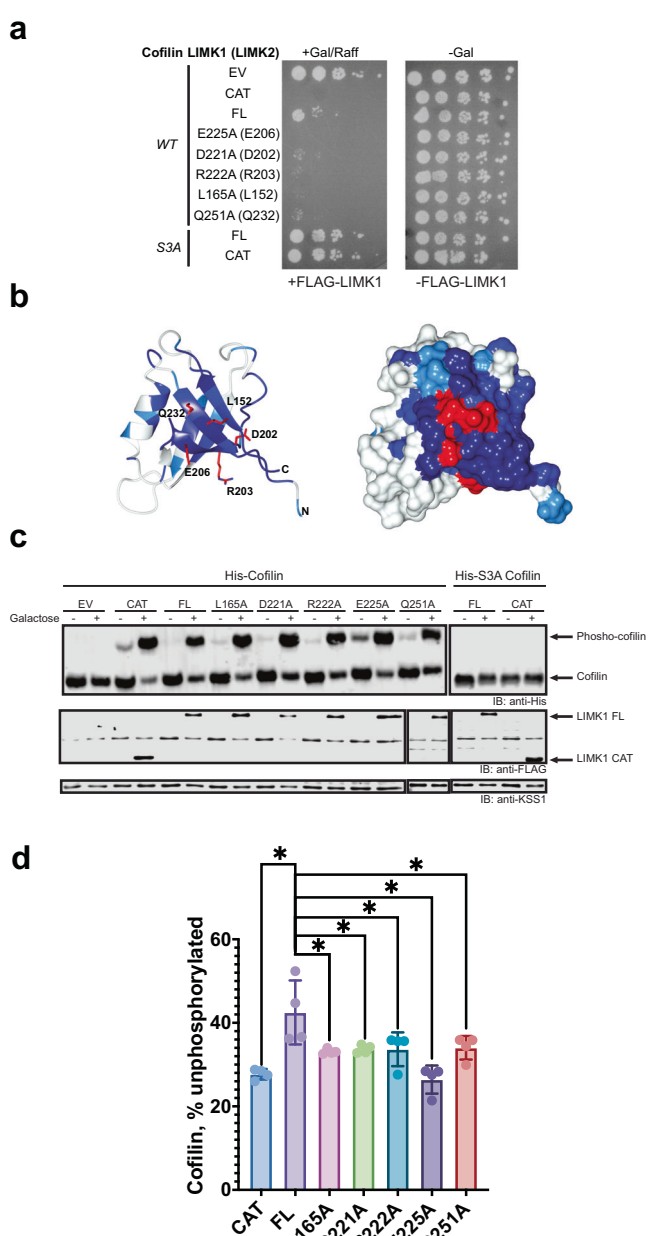

**Fig. 4 | PDZ domain mutants suppress yeast growth and increase cofilin phosphorylation. a** Serial dilutions of *cof1Δ* yeast expressing human cofilin and the indicated human LIMK1 mutants. Controls of human LIMK1 constructs, full-length (FL), kinase domain (CAT), unphosphorylatable cofilin S3A (S3A) and empty vector (EV). Mutants of full-length LIMK1: E225A, D221A, R222A, L165A, Q251A. Corresponding LIMK2 residue shown in parentheses. Five-fold dilutions of yeast cultures were plated on solid media in the presence of glucose (-Gal) or galactose and raffinose (+Gal/Raff) to induce LIMK1 expression. Plates were grown at 30 °C for 2 days (glucose plate) or 4 days (galactose plate). Representative of 3 independent experiments. **b** Mutants assessed are shown on the cartoon and surface representations of the conservation map of LIMK2 PDZ domain. Residues shown and equivalent human LIMK1 residue numbers: L152 (L165 in LIMK1), Q232 (Q251 in LIMK1), D202 (D221 in LIMK1), R203 (R222 in LIMK1) and E206 (E225 in LIMK1). **c** Immunoblots of lysates corresponding to yeast plated in (**a**). Cofilin species are separated based on phosphorylation state by Phos-tag SDS-PAGE. KSS1 serves as a loading control. Images are representative of $N = 4$. **d** Quantification of immunoblots measuring the percentage of total unphosphorylated cofilin. Statistical analysis was carried out using a non-parametric unpaired two-sided Mann-Whitney test. Data are mean values (bar graph) +/− SD (error bars), and individual measurements are plotted (dots, $N = 4$). One star (*) indicates $p < 0.05$ for all samples compared. *p*-values: FL vs CAT: $p = 0.0286$, FL vs L165A: $p = 0.0286$, FL vs D221A: $p = 0.0286$, FL vs R222A: $p = 0.0286$, FL vs E225A: $p = 0.0286$, FL vs Q251A: $p = 0.0286$. A total of 4 replicates were analyzed using GraphPad Prism.

Based on superposition of over 40 AlphaFold models of full-length LIMK1 and LIMK2 in different species, we found that the βA-βF-βD surface is almost completely surface exposed, with a small portion of the surface consistently found to interact with the adjacent LIM2 domain (residue L152 and residues of βA which makes an anti-parallel β-sheet interaction with the LIM2 domain). In these models, residue E206 (LIMK2) / E225 (LIMK1) is always surface exposed further supporting our finding that the βA-βF-βD surface has the potential to regulate the kinase, and also allows for an extended surface consistent with previous literature suggesting a role for the LIM domains in autoregulation[8,10,13,18,19]. The changes in kinase activity that we observe suggest that disruption of the surface that potentially mediates autoregulatory interactions between the PDZ domain and the kinase domain allows LIMK to reach a more "open" conformation. We interpret this to suggest multiple independent, or partially independent, steps are required to fully activate the LIM domain kinases, including both disruption of N-terminal domain interactions with the kinase domain and activation loop phosphorylation by upstream activators. A detailed biophysical exploration of these potential direct interactions is therefore warranted. Overall, our study clearly demonstrates that a previously unidentified surface on the PDZ domain plays a pivotal role in autoregulation of the LIM domain kinases.

region and in the βB-αB cleft do not impact activity. Mutation of these residues caused the isolated PDZ domain to be insoluble when expressed in bacteria, similar to some of the surface residues (Supplementary Fig. 8), suggesting that proper folding of the PDZ domain is required for autoregulation. It is important to note, however, that while our analyses provide a clear demonstration that the LIMK PDZ domain is important for changes in the activation loop phosphorylation of the LIM domain kinases and consequent changes in kinase activity, the work does not formally prove that this is mediated by a direct PDZ-kinase domain interaction. Although we consider this to be the likeliest possibility that results in changes in LIMK activation loop phosphorylation, other potential mechanisms include altered protein expression or stability, and changes to recognition of the LIMK as a substrate by upstream kinases or phosphatases. Notwithstanding these caveats, our studies demonstrate that a previously unidentified and completely conserved surface on the properly folded PDZ domain is required for normal autoregulation of the LIMKs.

This work provides a molecular level insight into the molecular surfaces important for autoregulation of the LIM domain kinases.

## Methods

### Protein expression and purification

The sequence encoding full-length human LIMK2 protein (UniProt ID: P53667) PDZ domain (131-25) was inserted using restriction enzymes BamHI and EcoRI into a modified *E. coli* expression vector pET28a containing an N-terminal FLAG tag followed by a (His₆) tag and a recognition sequence for tobacco etch virus (TEV) protease. A C173S point mutation was introduced using QuikChange Lightning site-directed mutagenesis kit (Agilent) to inhibit the disulfide bond formation and improve stability for crystallization experiments. Primers shown in Supplementary Table 1. Solubility testing of PDZ domain mutants was conducted on a C173S mutant background.

His tagged LIMK2 PDZ was expressed in BL21(DE3) cells (Millipore Sigma) by induction with 0.5 mM isopropyl β-D-thiogalactopyranoside (IPTG) overnight at 16 °C. Cells were harvested by centrifugation at 2000 × g and lysed by suspension in nickel binding buffer (50 mM HEPES pH 7.5, 500 mM NaCl) including of 0.1 M phenylmethylsulfonyl fluoride (PMSF), Roche complete (EDTA-Free protease inhibitor tablet)

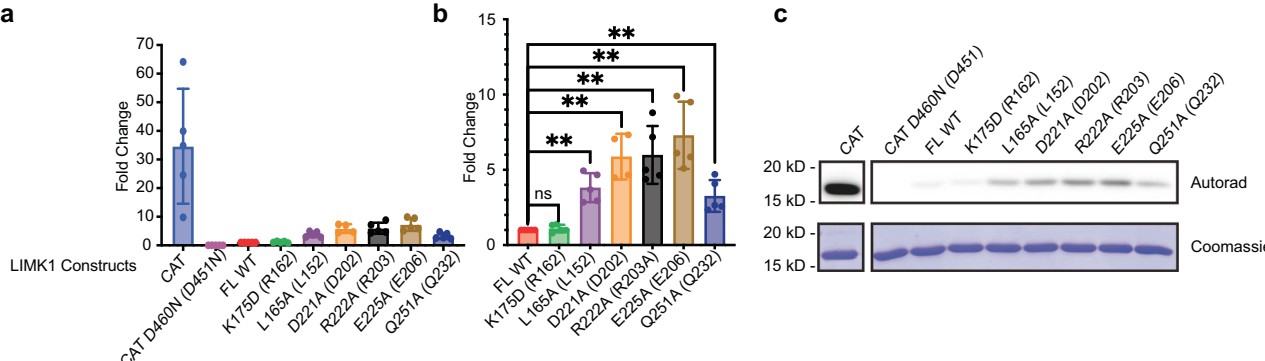

**Fig. 5 | Increased in vitro kinase activity for LIMK1 PDZ mutants.** Quantified autoradiography from radiolabel cofilin kinase activity of FLAG-LIMK1 constructs purified from yeast. **a** Full-length wild type (WT), kinase domain alone (CAT), and catalytically inactive kinase domain (CAT D460N) were used as positive and negative controls, respectively. PDZ domain mutants of conserved residues in FL LIMK1 are shown (equivalent residue in LIMK2 shown in parentheses). **b** Graph focused on full-length mutant constructs compared to WT. **c** Lower panel shows representative autoradiography reading of cofilin phosphorylation for kinase assays with corresponding Coomassie staining. Statistical analysis was carried out using a non-parametric unpaired two-sided Mann-Whitney test. Data are mean values (bar graph) +/− SD (error bars), and individual measurements are plotted (dots, $N = 5$). Two stars (**) indicates $p < 0.005$ for all samples compared. $p$-values: FL vs CAT: $p = 0.0079$, FL vs L165A: $p = 0.0079$, FL vs D221A: $p = 0.0079$, FL vs R222A: $p = 0.0079$, FL vs E225A: $p = 0.0079$, FL vs Q251A: $p = 0.0079$, FL vs K175D: $p = 0.6825$. A total of 5 replicates (4 for D221A) were analyzed using GraphPad Prism.

and lysozyme, followed by freeze/thaw cycles and sonication. Lysates were clarified by centrifugation at 5000 × g for 1 h. Supernatant was applied to nickel beads for affinity purification (Ni Sepharose 6 Fast Flow, GE Healthcare). Following elution of bound proteins by increasing concentrations of imidazole in nickel-binding buffer, the His tag was removed from PDZ by incubation with TEV protease overnight during dialysis against buffer containing 50 mM HEPES pH 7.5, 500 mM NaCl. The cleavage reaction was then flowed over a nickel affinity column (HisTrap Fast Flow, GE Healthcare) to remove the liberated His tag, uncleaved His-tagged protein and the His-tagged TEV protease. The flow-through containing untagged PDZ protein was concentrated in a centrifugal filter (Amicon Ultra, Millipore Sigma), diluted to a salt concentration of 37 mM NaCl, and applied to a 5 ml anion exchange column (Mono Q GE Healthcare) equilibrated in 20 mM Tris pH 7.5 buffer. Protein was eluted with a continuous gradient of NaCl, ranging from 0% to 40% 1 M NaCl, and 20 mM Tris pH 8, with the protein eluting at 12% 1 M NaCl. The eluted peak was concentrated and then purified by size exclusion chromatography on a Superdex 75 10/300 GL. LIMK2 PDZ eluted as a monodisperse peak.

### Yeast protein expression

The *cof1Δ* yeast strains co-transformed with pRS423 GPD-S3A His-Cofilin and FLAG-LIMK1 expression constructs were grown from an individual colony overnight at 30 °C in 5 mL of SC-His-Leu with 2% raffinose. The next day the culture was diluted into 500 ml SC-His-Leu with 2% raffinose to an $OD_{600}$ of 0.1 and grown to an $OD_{600}$ of ~2. Next, 225 ml of 3.5x yeast extract, peptone solution (YP) and 80.5 mL of 10% galactose were added to the flask to induce expression of LIMK1 for 8 h. Yeast were centrifuged at 2600 × g for 30 min at 4 °C. Cells were resuspended in 10 ml of sterile water, repelleted, snap frozen in liquid nitrogen, and stored at −80 °C.

FLAG- LIMK cell pellets were thawed on ice and resuspended in 5 ml of FLAG lysis buffer (50 mM HEPES, pH 7.4, 150 mM NaCl, 1 mM EDTA, 0.5% Triton X-100, 10% glycerol, 0.5 mM DTT, 1 mM PMSF, 2 µg/mL pepstatin A, 2.5 mM $NaPP_i$, 1 mM βGP, 1 mM $Na_3VO_4$ and Roche complete EDTA-Free protease inhibitor tablet). Resuspended pellets were distributed into 10 microtubes each containing 150 µl of glass beads and lysed by agitating the beads with a vortex mixer. Lysates were transferred to fresh tubes and centrifuged at 800 x g for 10 min at 4 °C. Thermo Scientific Pierce anti-DYKDDDDK M2 resin (300 µL) equilibrated in lysis buffer was added to the supernatant and incubated

with rotation for 2 h at 4 °C. The resin was pelleted (197x g, 2 min, 4 °C), resuspended in 1 ml lysis buffer, and washed twice with wash buffer (50 mM HEPES, pH 7.4, 100 mM NaCl, 1 mM DTT, 1 mM βGP, 100 µM $Na_3VO_4$, 0.01% NP-40, 10% glycerol). FLAG elution buffer (400 µl of 50 mM HEPES, pH 7.4, 100 mM NaCl, 1 mM DTT, 1 mM βGP, 100 µM $Na_3VO_4$, 0.01% NP-40, 10% glycerol and 0.5 mg/ml of FLAG peptide) was added and the resin was incubated at 4 °C while rotating for 2 h. Resin was then centrifuged at 197x g for 2 min, and eluted protein was collected, aliquoted, snap-frozen in liquid nitrogen and stored at −80 °C. Purity and protein concentration were estimated against a BSA standard curve on SDS-PAGE (15% acrylamide) with Coomassie staining.

### Crystallization, data collection and structure determination of LIMK2 PDZ domain

Initial small cube-like clusters of PDZ crystals were obtained by sparse matrix screening using a TTP Labtech Mosquito by vapor diffusion in sitting drops 4°C with a 2:1 (v/v) ratio of purified protein to reservoir solution containing 0.1 M HEPES pH 7.5, 10% 2-propanol and 20% PEG 4000. Optimization of crystals was carried by using sitting drop methodology. Crystals were harvested from the drop, quickly incubated in 15% glycerol as a cryoprotectant and flash-cooled in liquid nitrogen. Four sets of diffraction data were collected from a single crystal at Northeastern Collaborative Access Team (NE-CAT) Beamline 24-ID-E at Argonne National Laboratory Advanced Photon Source, processed using XDS[63], and scaled using SCALA[64]. The data were processed in space group $P21$, with unit cell dimensions $a = 80.9$ Å, $b = 83.0$, $c = 83.1$ Å, $α = 90°$, $β = 96.6°$, $γ = 90°$. Matthew's probability calculation indicated 8 copies of the PDZ domain in the asymmetric unit. Phaser[65] confirmed the prediction using the predicted AlphaFold structure of LIMK2 PDZ as model (residues 131-250, LIMK2-AF-P53671-F1-model_v2.pdb). Model building was performed in Phenix Autobuild[66], and manual autobuilding in Coot[67] was performed. Refinement was carried out in Phenix refine[68].

### Conservation analysis

LIM kinase 1 and 2 sequences were identified using NCBI BLAST[69]. A total of 421 sequences were aligned using the Clustal Omega[70] server and visualized using JalView[71]. PDZ sequences from other proteins were identified using NCBI BLAST[69]. For PDZ containing human proteins, a total of 967 sequences were aligned. Sequences were aligned using the Clustal Omega[70] server and visualized using JalView[71].

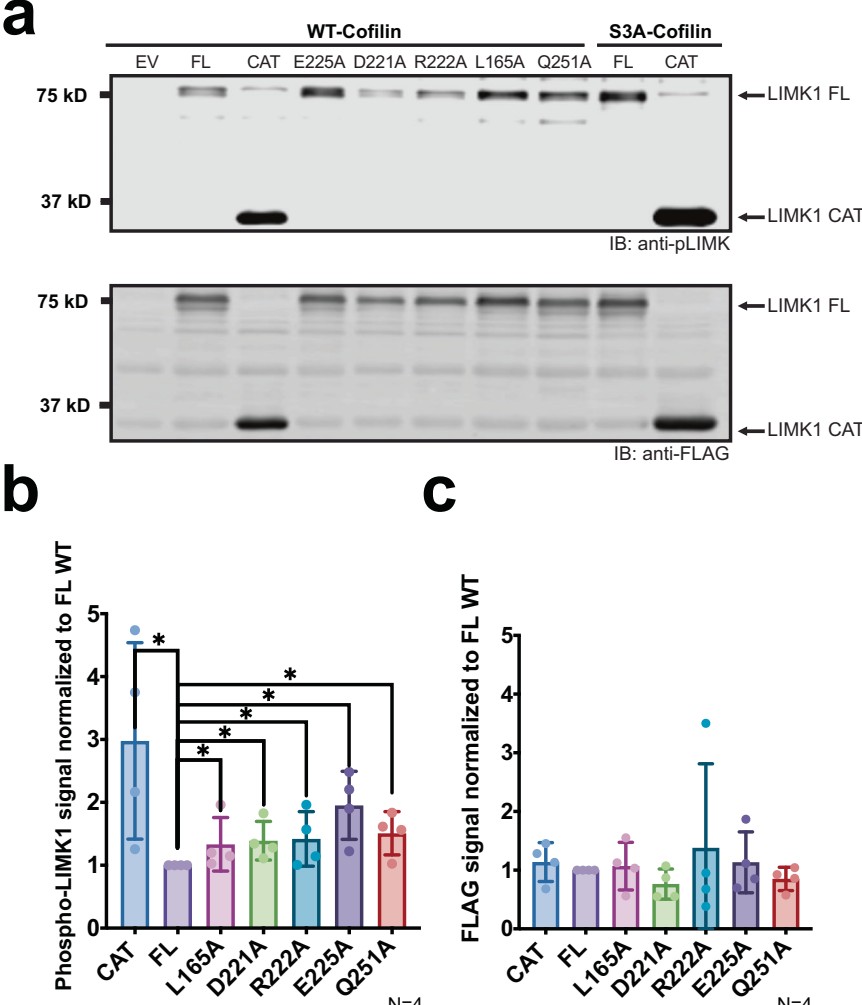

**Fig. 6 | Assessment of LIMK expression and activation loop phosphorylation in yeast. a** Western Blot assessing activation loop phosphorylation in LIMK1 constructs from our yeast growth assays. The top panel shows yeast lysates blotted with anti-pLIMK antibody and the bottom panel purified yeast protein blotted with anti-FLAG. CAT indicates catalytic domain, FL indicates full-length LIMK1. Mutations of full-length LIMK1 are indicated. **b** Quantification of Western blot signal. For each construct, phospho-LIMK1 signal was normalized to FLAG signal then compared to full-length LIMK1 (FL). **c** Quantification of FLAG signal in all four replicates, normalized to FLAG-LIMK1 FL signal. All mutant constructs signals are non-significantly different from each other. Statistical analysis was carried out using a non-parametric unpaired two-sided Mann-Whitney test. Data are mean values (bar graph) +/− SD (error bars), and individual measurements are plotted (dots, $N = 4$). $p$-values: FL vs CAT: $p = 0.0143$, FL vs L165A: $p = 0.0143$, FL vs D221A: $p = 0.0143$, FL vs R222A: $p = 0.0143$, FL vs E225A: $p = 0.0143$, FL vs Q251A: $p = 0.0143$. One star (*) indicates $p < 0.05$. A total of 4 replicates were analyzed using GraphPad Prism.

## Yeast growth assays

The high copy vector for constitutive expression of N-terminally His$_6$-tagged human cofilin-1 in yeast (pRS423-GPD-cofilin-1), and the galactose-inducible expression vector for N-terminally FLAG epitope-tagged LIMK1 $_{catalytic kinase domain (pRS415-GAL-LIMK1-CAT)}$ was cloned into the BamHI and XhoI sites of pRS415-GAL were previously described[11]. All point mutants were prepared by using QuikChange Lightning site-directed mutagenesis kit (Agilent) and verified by sequencing through the entire open reading frame. Yeast expressing human cofilin-1 were generated by plasmid shuffle starting with a *cof1Δ* strain supported by expression of yeast Cof1 from a *CEN URA3* plasmid (MHY8282, obtained from Mark Hochstrasser's laboratory[72]). This strain was transformed with pRS423-GPD-cofilin-1 (WT or S3A mutant), and then the yeast Cof1 plasmid was evicted by selection on solid media containing 5-FOA. This strain was then transformed with and the indicated LIMK1 expressing plasmids or the corresponding empty vectors. To assess the impact of LIMK1 expression on cell growth, yeast were grown overnight at 30 °C in synthetic complete media lacking histidine

and leucine (SC-His-Leu) containing 2% glucose. The following day, cultures were diluted into SC-His-Leu containing 2% raffinose and grown overnight to mid-log phase. Serial 5-fold dilutions (starting OD = 0.2) were then spotted onto SC-His-Leu agar plates containing either 2% glucose or 2% raffinose/1% galactose, and plates were incubated at 30 °C until colonies were visible at the highest dilution of the empty vector strain. Point mutations in pRS415-GAL-FLAG-LIMK1 were introduced substituting residues Leu165, Asp221, Arg222, Glu225, Gln251 with alanine, Lys175 with aspartate, and Thr508 with two glutamic acids using QuikChange Lightning site-directed mutagenesis kit (Agilent). Primers used for mutagenesis are listed in Supplementary Table 1.

## Immunoblotting

Yeast cultures (500 ml) were grown to an $OD_{600}$ of 1–2 in 2% raffinose at 30 °C, and then 1% galactose was added to induce LIMK1 expression. After 4 h, cells were harvested and lysed using a TCA extraction protocol adapted from[73] with the following modifications. Yeast cells were

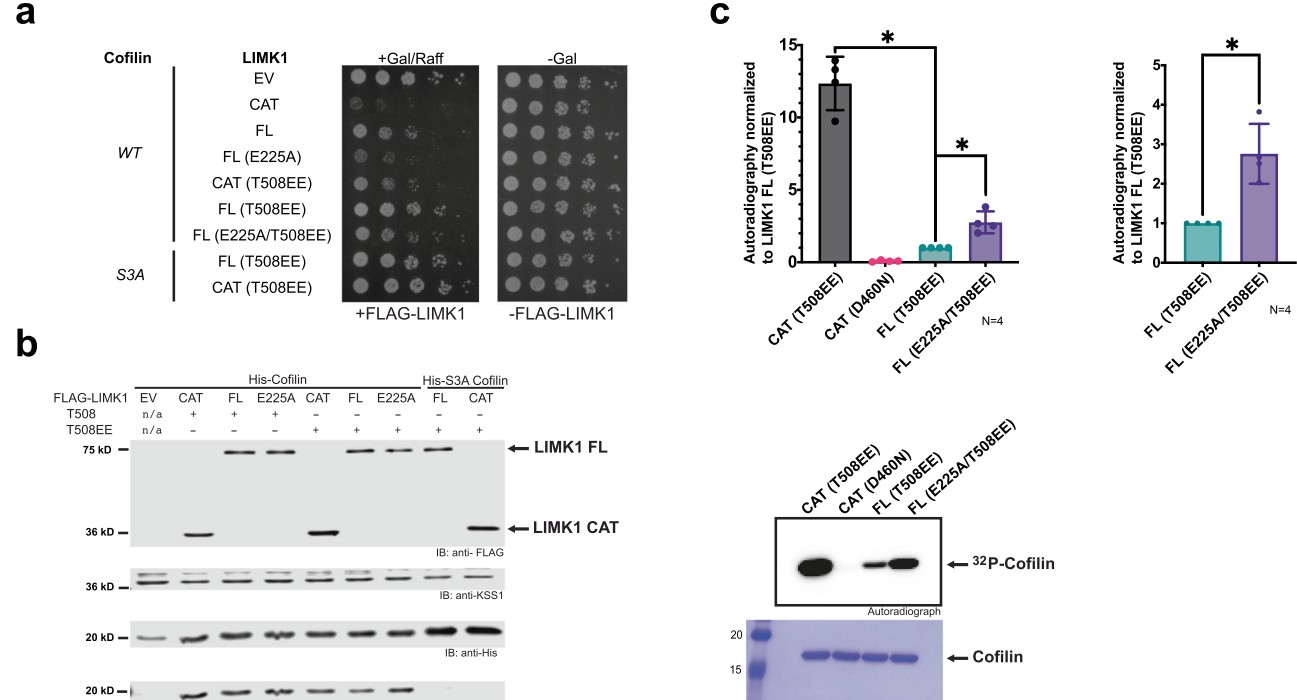

**Fig. 7 | Impact of LIMK1 activation loop and PDZ domain mutations on activity.**
**a** Serial dilutions of *cof1Δ* yeast expressing human cofilin and the indicated human LIMK1 and constitutively active mutants. Controls of human LIMK1 constructs, full-length (FL), kinase domain (CAT) and empty vector (EV); controls of unphosphorylatable cofilin S3A (S3A). Activation loop mutants indicated by T508EE, PDZ mutants indicated by E225A. Five-fold dilutions of yeast cultures were plated on solid media in the presence of glucose (-Gal) or galactose and raffinose (+Gal/Raff) to induce LIMK1 expression. Plates were grown at 30 °C for 2 days (glucose plate) or 4 days (galactose plate). Representative of 3 independent experiments.
**b** Immunoblot analysis of FLAG-LIMK1 constructs expressed in yeast. Blots for FLAG, KSS1 loading control, His-cofilin (anti-His), and cofilin phospho-Ser3 (anti-p-Cofilin). FL indicates full-length LIMK1, and CAT indicates LIMK1 catalytic domain. Mutants as indicated. **c** Quantified autoradiography of LIMK1 kinase activity

towards cofilin. Activity towards cofilin monitored for full-length T508EE mutant LIMK1 (FL T508EE), T508EE mutant kinase domain alone (CAT T508EE), catalytically inactive kinase domain (CAT D460N) and full-length PDZ E225A and activation loop T508EE mutant LIMK1 (FL E225A/T508EE). Equivalent LIMK1 and LIMK2 residues, respectively: E225/E206, D460/D451, T508/T505. Right graph shows only on full-length LIMK1 T508EE and full-length LIMK1 E225A/T508EE. Left lower panel shows representative autoradiography reading of cofilin phosphorylation for kinase assays and corresponding Coomassie staining. Statistical analysis was carried out using a non-parametric unpaired two-sided Mann-Whitney test. Data are mean values (bar graph) +/− SD (error bars), and individual measurements are plotted (dots, $N = 4$). One star (*) indicates $p < 0.05$ for all samples compared. *p*-values: FL T508EE vs CAT T508EE: $p = 0.0286$, FL T508EE vs FL E225A/T508EEA: $p = 0.0286$. A total of 4 replicates were analyzed using GraphPad Prism.

resuspended in a lysis buffer containing 10% TCA, 25 mM $NH_4OAc$, 10 mM Tris HCl, pH 8.0, and 1 mM DTT. Glass beads were added to the resuspended lysate and vortexed for 5 min at 4 °C. Lysed cells were centrifuged at 16900 x g in a 4 °C centrifuge for 10 min. Pellets were resuspended in 0.1 M Tris pH 11 and 3% SDS. Pellets containing precipitated proteins were diluted 1:10 and then used for BCA assays. BCA assays were used to normalize the amount of protein added. Equal amounts of lysate with 4X SDS-PAGE loading buffer (7 μg per lane) were fractionated by SDS-PAGE and transferred to polyvinyl difluoride (PVDF) (Sigma, IPFL85R) membrane. Membranes were blocked in Tris buffer saline (TBS) with 5% non-fat milk for 1 h and probed overnight at 4 °C with the indicated primary antibodies: mouse anti-FLAG antibody (Sigma, #F3165,1:5,000 dilution). Membranes were incubated for 30 min in fluorescently labeled secondary antibodies IRDye® 800CW goat anti-mouse IgG secondary Antibody (Licor, #D10603-05) and goat anti-rabbit IgG (H + L) Highly Cross-Adsorbed Secondary Antibody, Alexa Fluor 680 (Invitrogen, #A21109) in 1:10,000 dilution in TBS with 5% bovine serum albumin (BSA) and 0.1% Tween20. Membranes were scanned using a Li-Cor Odyssey Imaging system. For the assessment of activation loop phosphorylation, 3.3 μg of FLAG-LIMK1 preparations purified from yeast were analyzed in the same manner. The following primary antibodies were used: mouse anti-FLAG antibody (Sigma, #F3165,1:5,000 dilution), rabbit anti-KSS1 (Santa Cruz Biotechnology, # sc-6775-R, 1:5,000 dilution), penta-His (Qiagen, # 34650, 1:5000), and p-Cofilin (Serine3) (Cell Signaling, #3311 S,

1;1000), phospho-LIMK1/LIMK2 antibody (Thr508/Thr505) (Cell Signaling, #3841 S 1;1000).

### Phos-tag SDS-PAGE analysis

To assess the impact of LIMK1 expression cofilin phosphorylation, yeast were grown overnight at 30 °C in SC-His-Leu containing 2% glucose. The following day, cultures were diluted into SC-His-Leu containing 2% raffinose and split into two groups. We added 1% galactose to one group to induce LIMK1 expression, and we added 1% glucose to the other group to repress LIMK1 expression. After 2 h, cells were harvested and lysed by TCA extraction as described above. Protein concentrations were determined by BCA assay, and 3 μg of each was fractionated using Phos-tag gels (SuperSep Phos-Tag 12.5% Cat #195-17991, and 7.5% Cat #192-18001, FUJIFILM Wako Chemicals). After fractionation, gels were incubated twice for 15 min in transfer buffer containing 10 mM EDTA to remove zinc ions before transfer to PVDF (Sigma, IPFL85R) membrane. Membranes were blocked and probed as described above with anti-Penta-His (QIAGEN, #34650) as the primary antibody.

### Mutagenesis and solubility test of His tagged LIMK2 PDZ mutants

Primers used are indicated in Supplementary Table 1. All mutants were expressed in BL21 cells. Overnight cultures were inoculated into 1 L ml of Luria broth, and protein expression was induced with isopropyl 1-

thio-β-d-galactopyranoside when $OD_{600}$ = 0.6. Cells were grown overnight at 18 °C, harvested, and resuspended in 10 ml of 500 mM NaCl and 20 mm Tris, pH 8.0, supplemented with DTT, protease inhibitors, lysozyme, and DNase I. Resuspended cells were lysed by three freeze/thaw cycles in a dry ice/ethanol bath followed by sonication. Lysates (100 µl) were centrifuged at 20400 × g for 10 min. The supernatants were separated from the pellets. Pellets were resuspended in 100 µl of 6 M urea and diluted two-fold in lysis buffer. Samples were run on a 15% acrylamide SDS-PAGE and visualized by Coomassie staining.

## Radiolabel kinase assays

Human cofilin was purified as previously described[11]. Kinase reactions (25 µl) contained 5 nM purified LIMK1 and 6.7 µM cofilin in 50 mM HEPES, pH 7.5, 100 mM NaCl, 5 mM $MgCl_2$, 5 mM $MnCl_2$, 20 µM ATP, 1 mM DTT, 0.1 µCi/ml $^{32}$P-ATP. Reactions were incubated 10 min at 30 °C, quenched by adding 1x SDS-loading buffer, and resolved by SDS-PAGE on a 15% polyacrylamide gel. Dried gels were exposed to a phosphor screen, and the level of phosphorylated cofilin was evaluated on a Bio-Rad Molecular Imager Fx system using Quantity One 1D Analysis software (Life Sciences Research). Data from 5 separate experiments were normalized to FLAG-FL LIMK1 signal, and statistical analysis was carried out using a non-parametric unpaired Mann-Whitney test in GraphPad Prism.

## Reporting summary

Further information on research design is available in the Nature Portfolio Reporting Summary linked to this article.

## Data availability

Coordinates and structure factors have been deposited in the Protein Data Bank under accession code 8GI4. X-ray diffraction images are available online at SBGrid Data Bank [https://doi.org/10.15785/SBGRID/1010]. Previously determined structures used in our analysis were obtained from the Protein Data Bank: 3EGG (spinophilin, PDZ), 5HEY (disk large homolog 4 PDZ), 3K1R (harmonin PDZ), 5G1E (syntenin-1 PDZ). The AlphaFold model of Human LIMK2 (AF-P53671-F1-model_v2.pdb) was obtained from the Alphafold Structure Database: https://alphafold.ebi.ac.uk/files/AF-P53671-F1-model_v2.pdb. The source data underlying Fig. 4d; 5a, b; 6b, c; 7c are provided as a Source Data file. Source data are provided with this paper.

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

## Acknowledgements

We thank Stephanie Hamill, Amy Stiegler Wyler, and Jaylissa Torres-Robles for helpful discussions. We thank Jim Murphy and Jaylissa Torres

Robles for training and support, and Mark Hochstrasser for provision of yeast strains. We thank Hua Jane Lou and Cameron Schmitz for their preliminary work testing LIMK1 activity with yeast growth assays. We thank the beamline scientists of NE-CAT at the Advance Photon Source. This work is based upon research conducted at the Northeastern Collaborative Access Team beamlines, which are funded by the National Institute of General Medical Sciences from the National Institutes of Health (P30 GM124165). The Eiger 16 M detector on the 24-ID-E beam line is funded by a NIH-ORIP HEI grant (S10OD021527). This research used resources of the Advanced Photon Source, a U.S. Department of Energy (DOE) Office of Science User Facility operated for the DOE Office of Science by Argonne National Laboratory under Contract No. DE-AC02-06CH11357. G.C.S. and J.A.S. were supported by National Institutes of Health training grant T32GM007324. G.C.S. supported by American Heart Association grants 835293 and 23DIVSUP1058562. This research was supported by R01GM102262 to B.E.T. and T.J.B.

## Author contributions

Conceptualization, Methodology, Writing: G.C.S, J.A.S., B.E.T. and T.J.B. Investigation, Data Curation, Visualization: G.C.S. Supervision: T.J.B.

## Competing interests

The authors declare no competing interests.
