## [Peer Review File · Nature Communications]

REVIEWER COMMENTS

Reviewer #1 (Remarks to the Author):

In their manuscript 'Autoregulation of the LIM kinases by their PDZ domain', Casanova-Sepúlveda et al. investigate a potential role of the LIMK PDZ domain in regulating the LIMK kinase activity. They solved a crystal structure of the LIMK2 PDZ domain. Based on LIMK sequence alignments, they identified conserved residues in the PDZ domain. Several of these residues were then mutated in the FL LIMK1 protein. The variants were finally probed in a kinase activity assay and in a previously established yeast viability assay.

General comments:

i Depending on the experiment, either LIMK1 (activity assay, yeast viability assay) or LIMK2 (PDZ domain structure model) were applied. Please explain why the respective isoform was chosen. And please comment on how the other isoform would have performed.

ii Mutating the PDZ domain in FL LIMK1 had an impact on the expression level as demonstrated in Figure 6A. Furthermore, the subcellular localization may differ between the variants, making it more or less likely for them to be co-localized with cofilin. And finally, the variants were differently phosphorylated/activated by upstream kinases as again demonstrated in Figure 6A. There are several possibilities of how mutating the LIMK PDZ domain can lead to changes in cofilin phosphorylation and yeast viability. A physical interaction between the LIMK PDZ and kinase domains is only one of them. Please discuss this accordingly.

iii Phosphorylation in T508 switches the LIMK1 catalytic domain from inactive to active. Is this also valid for the FL LIMK1? Could you please hypothesize on the interplay between T508 phosphorylation and PDZ autoregulation?

More specific comments:

Line 61. It may make sense to mention TESK1/2 here.

Line 75. Please explain the difference between the LIMK 'autoinhibited state' and the LIMK 'inactive state'.

Line 122. Please discuss why these boundaries were chosen. Was the LIMK2 AlphaFold model considered? Starting from D120 would have left the beta sheet intact. And overall, the LIMK LIM2 and PDZ domains seem to have merged into one single domain - this may also explain the unique sequence features of the LIMK PDZ domain.

Line 191. Please discuss whether the differences in yeast viability between CAT and WT can rely on differences in the T508 phosphorylation grade.

Line 205. In Figure S5, all LIMK1 variants appear to have similar expression levels. Is this in contrast to Figure 6A? And surprisingly, the kinase activities of all variants appear to be similar, with no sign of autoinhibition via the PDZ domain.

Line 213. Is it the low activity of non-phospho LIMK1 that is assessed here? Or is it the activity of pT508 LIMK1? The variant T508A would have been helpful. Please indicate whether or not the kinase activities correlates with the T508 phosphogrades.

Line 224. Is this autophosphorylation? Also the D460N variant is phosphorylated in Figure S7. Is it rather yeast kinases phosphorylating LIMK1?

Line 282. Another conclusion from the AlphaFold model is the high-confidence placement of the LIM2 domain close to the conserved PDZ surface patch. Furthermore, in reference 19, the N-terminal LIM domains (and not the PDZ domain) are suggested to be the key regulators of LIMK kinase activity. How can this be followed up in the future?

Line 305. Several statements in this paragraph strongly suggest that the LIMK PDZ and kinase domains interacted physically. Please consider modifying the wording or provide additional experimental evidence for this claim.

Reviewer #2 (Remarks to the Author):

In their manuscript 'Autoregulation of the LIM kinases by their PDZ domain', the crystal structure of human LIMK2 PDZ domain was determined, with a shallow β B- α B binding groove, which may explain why the LIMK PDZ domains, unlike canonical PDZ domains, cannot interact with the carboxy-terminal peptides. Based on the sequence conservation and electrostatic potential analysis, the authors defined a

hydrophobic patch, the β A- β F- β D surface, conserved in all LIMK PDZ domains. Furthermore, they used human LIMK1 proteins ectopically expressed in yeast to examine the importance of this β A- β F- β D surface on LIMK1 activity. This study provides insights into the autoregulation mechanism of the LIM kinases by a previously unidentified surface on the PDZ domain. However, revision of the manuscript to address the concerns below will make it more relevant to the broad readership of Nature Communications.

1. PDZ domains generally recognize the C-terminal residues of binding partners, while the LIMK PDZ domain likely does not interact with C-terminal peptides. The authors mentioned that potential binding partners of the LIMK PDZ domain have not been identified (line 91). The N-terminal LIM domains of LIMK were reported to interact with the C-terminal kinase domain and negatively regulate the kinase activity (ref. 18 and 19). Would the PDZ domain also interact with the kinase domain (and/or the LIM domains)? It would be helpful to carry out interaction analysis among the LIMK PDZ, LIM1/2 and kinase domains, as well as trans-inhibition assay of the PDZ domain and the N-terminal fragment containing both LIM and PDZ domains (WT and PDZ mutations) on the activity of kinase domain.

2. There are more than 250 PDZ domains in over 150 human proteins, which could bind short linear peptides or non-peptide ligands. The authors found the LIMK PDZ domain adopting a canonical PDZ fold with an unusual non-canonical mechanism for peptide/partner recognition. Please introduce the structural features/differences of canonical and non-canonical PDZ domains, as well as the different binding modes (PDZ domains can mediate protein-protein interactions through alternate modes or alternative binding surfaces). The structure of the human LIMK2 PDZ domain contains 8 copies per asymmetric unit. What is the oligomeric state of the human LIMK2 PDZ domain in solution? Does the crystal packing give potential binding/recognition mode(s) of LIMK PDZ domain and partner?

3. The 'G-L-G-F' or 'X- β -G- β ' motif was first mentioned without a detailed description (line 117).

Canonical PDZ domains utilize the β B- α B groove to bind partner peptides and terminal carboxylate groups, while the inward-facing arginine residue of the human LIMK2 PDZ domain encroaches on the expected carboxylate binding site resulting in a shallow binding groove. Please provide a structural comparison of the human LIMK2 PDZ domain and a canonical PDZ domain bound with a peptide to illustrate the significance of this Arg residue.

4. The newly-identified β A- β F- β D surface is conserved among all LIMK orthologs. The authors generated point mutations of this surface on human LIMK1, instead of human LIMK2, to examine its importance? The PDZ domains of human LIMK1 and LIMK2 are highly conserved. How are two full-length proteins, sequence conserved and function redundantly? The β A- β F- β D surface is hydrophobic (line 169 and Figure 3B). Why the point mutations used (L165, E225, D221, R222 and Q251) involve only one hydrophobic residue?

In line 204, human LIMK1 E225A mutations resulted in a complete loss of yeast growth, suggesting that the activity of this mutant LIMK1 comparable to the uninhibited kinase domain alone. As shown in Figure S5, the phosphorylation levels of p-Cofilin in yeast expressing LIMK1 WT and PDZ mutants are almost the same. The catalytic domain and LIMK1-WT were expected to have a dramatic difference. Moreover, the result of in vitro kinase activity assay showed that the E225A mutant of LIMK1 FL increased ~7.5 fold in kinase activity, while the catalytic domain showed a ~35-fold increase compared to FL WT. This result indicates that E225A mutation on the PDZ domain is not sufficient to relieve the autoinhibition of LIMK. Why did the results of in vitro kinase assay and yeast growth assay not match well?

Minor points:

☒ The full name of LIM?

☒ In line 195, it should be Figure S5, not Figure 4A.

☒ In line 213, 'most mutants showed an approximate 4-fold increase in kinase activity, with E225A demonstrating a ~7.5-fold increase.' As shown in Figure 5, mutants D221A and R222A showed comparable changes to E225A.

☒ The authors examined the phosphorylation of the activation loop of human LIMK WT and PDZ mutants using yeast lysates (Figure 6A) and purified protein from yeast (Figure S7). However, the results of D221A and R222A in Figure 6A and Figure S7 are inconsistent.

☒ In Figure 6A, the top panel used yeast lysates to show the phosphorylation of LIMK1 while the second panel used purified protein to show the protein expression according to the figure legend. And, Q232A is LIMK2 mutant, equivalent to LIMK1 Q251A mutant.

☒ Figure S6, the figure and legend are inconsistent.

☒ Line 249, 'reveals an unusual addition to this common fold'?

Response to Reviewers

Reviewer #1 (Remarks to the Author):

In their manuscript 'Autoregulation of the LIM kinases by their PDZ domain', Casanova-Sepúlveda et al. investigate a potential role of the LIMK PDZ domain in regulating the LIMK kinase activity. They solved a crystal structure of the LIMK2 PDZ domain. Based on LIMK sequence alignments, they identified conserved residues in the PDZ domain. Several of these residues were then mutated in the FL LIMK1 protein. The variants were finally probed in a kinase activity assay and in a previously established yeast viability assay.

>RESPONSE: We thank the reviewer for their thoughtful and insightful comments. We have addressed each of their queries and comments in detail below.

General comments:

i Depending on the experiment, either LIMK1 (activity assay, yeast viability assay) or LIMK2 (PDZ domain structure model) were applied. Please explain why the respective isoform was chosen. And please comment on how the other isoform would have performed.

>RESPONSE: We thank the reviewer for highlighting this aspect of our study which we realize was not adequately explained in our original manuscript. We began this project focusing exclusively on LIMK1. We extensively attempted to crystallize the PDZ domain of LIMK1 using a variety of constructs from different species, ranging from humans to *Drosophila*. However, the PDZ domain of LIMK1 was recalcitrant to crystallization. We therefore conducted detailed sequence analysis of LIMK proteins across 421 species and observed that the PDZ domains of LIMK1 and LIMK2 are highly similar (for example, the human LIMK1 and LIMK2 PDZ domains are 47% identical and 81% similar). We therefore decided to attempt crystallization of the LIMK2 PDZ domain from a range of species. We found that the human LIMK2 PDZ domain crystallized and this allowed structure determination. Importantly, our conservation analysis of the PDZ domain demonstrated a completely conserved surface on this structure between LIMK1 and LIMK2, and therefore we focused on this surface for our mutagenesis studies. Our assumption, based on the extensive sequence similarity of this surface among the LIMKs, was that LIMK1 and LIMK2 should function in a similar manner. However, the reviewer raises an important question

regarding whether this is a valid assumption to make, and whether the LIMKs do indeed function similarly across the isoforms.

To address this query, we conducted growth assays of yeast expressing human LIMK2 analogous to those in our manuscript involving LIMK1. We have assessed both full-length and catalytic domain of LIMK2 and introduced the E206A mutation into the full-length construct (equivalent to E225A in LIMK1). Surprisingly, we found that none of the LIMK2 constructs, even the catalytic domain alone, affected yeast growth. We find that human cofilin is phosphorylated at low levels and notably, phosphorylation of human cofilin in these assays is appreciably lower than when using LIMK1 (**Reviewer Figure 1**) (compared to experiments presented and discussed in **Figure 4** below). In our experience growth suppression only occurs at a very high level of cofilin phosphorylation, likely because a small pool of unphosphorylated cofilin is sufficient to support growth. The lower level of cofilin phosphorylation (~40-50% of total) induced by LIMK2 in yeast is thus insufficient to produce a growth phenotype. The reason for reduced LIMK2 activity in yeast compared to LIMK1 is not immediately clear, and potentially could be associated with the ability of upstream kinases to act on this isoform (yeast have no LIMKs, so all upstream kinase-mediated phosphorylation is 'off-target'). We present **Reviewer Figure R1** below to describe these results.

Reviewer Figure R1. Phosphorylation of cofilin by LIMK2 expression in yeast. Reconstitution of mammalian LIMK2-cofilin pathway in yeast yields low phosphorylation of cofilin after 4 hours. **a)** Yeast growth assays using LIMK2, equivalent residue in human LIMK1 is shown in parentheses. **b)** Immunoblot experiments to test the expression of LIMK2 in yeast. Expression of Flag-LIMK2 is stable, and results in activation loop phosphorylation of LIMK2 constructs. **c)** Phos-tag gel separating phosphorylated cofilin and unphosphorylated cofilin from yeast growth experiments.

ii Mutating the PDZ domain in FL LIMK1 had an impact on the expression level as demonstrated in Figure 6A.

>RESPONSE: The reviewer highlighted the expression levels in **Figure 6a**. We agree with the reviewer that in the initial submission **Figure 6a** showed variations in expression levels and we thank the reviewer for pointing this out. To address this concern, we conducted further replicates of the experiments presented in **Figure 6a** (shown below). We find that the expression levels are more equal, although some differences in expression level remain. Phospho-LIMK1 (pThr508) is reported in **Figure 6** as normalized to FLAG levels. We have conducted these experiments four

times and find that the expression levels are consistent (each of the repeats are shown in **Reviewer Figure R2**, below). The source of the excessive variation in the original submission's **Figure 6a** may have been due to protein storage – the new analyses are conducted with fresh samples that had not been frozen, whereas the original samples had been stored in the freezer for an extended period prior to analysis.

Updated Figure 6: Assessment of LIMK expression and activation loop phosphorylation in yeast. a) Western Blot assessing activation loop phosphorylation in LIMK1 constructs from our yeast growth assays. The top panel shows yeast lysates blotted with anti-pLIMK antibody and the bottom panel purified yeast protein blotted with anti-FLAG. CAT indicates catalytic domain, FL indicates full-length LIMK1. Mutations of full-length LIMK1 are indicated. **b)** Quantification of

Western blot signal. For each construct, phospho-LIMK1 signal was normalized to FLAG signal then compared to full-length LIMK1 (FL). **c)** Quantification of FLAG signal in all four replicates, normalized to FLAG-LIMK1 FL signal. All mutant constructs signals are non-significantly different from each other. Statistical analysis was carried out using a non-parametric unpaired Mann-Whitney test. One star (*) indicates $p < 0.05$. A total of 4 replicates were analyzed using GraphPad Prism.

Reviewer Figure R2. We repeated the experiments for Figure 6 four times using non-frozen samples and found that expression levels are not significantly different when quantified and normalized to FLAG-LIMK1 FL WT, compared to all PDZ mutants 6A. **a)** Four replicates used in the quantifications. **b)** Quantification of FLAG signal in all four replicates, normalized to FLAG-LIMK1 WT signal. All mutant construct signals are not significantly different from each other. (Panel is also shown in updated Figure 6c).

Furthermore, the subcellular localization may differ between the variants, making it more or less likely for them to be co-localized with cofilin.

>RESPONSE: The reviewer raises the interesting possibility that differences in sub-cellular localization may provide a basis for differences in cofilin phosphorylation and cell growth. While conceptually we agree that LIMK mutations have the potential to alter its compartmentalization, we believe it is most likely differences in kinase activity rather than localization are responsible for effects on cell growth. First, in mammalian cells LIMK is established to be retained in the cytoplasm through a nuclear export signal (Yang and Mizuno, 1999, full reference below), so that mutations would if anything be expected to induce nuclear localization and *decrease*, rather than increase, cofilin phosphorylation. While theoretically mutations might promote LIMK accumulation at the cell periphery or to sites of actin polymerization, in practice this would be very difficult to assess. Based on its localization in mammalian cells, our use of a strong promoter that expresses LIMK to high levels, and our observation that the majority of cofilin molecules are phosphorylated upon LIMK expression, the bulk of LIMK likely localizes diffusely in the cytoplasm. In all likelihood, if mutations do change localization, it would involve a fraction the LIMK molecules, and on this background a small pool of relocalized LIMK would be difficult to detect. Finally, expression of hyperactive mutants is expected to affect cytoskeletal organization, which could itself affect LIMK localization within the cytoplasm. For these reasons, we were concerned that the substantial investment involved in setting up assays of LIMK localization (generating GFP-tagged expression constructs, optimizing growth and microscopy conditions, etc.) would be unlikely to be informative. We therefore focused our efforts on other experiments that addressed more directly mechanisms of LIMK regulation.

Yang, N., Mizuno, K. Nuclear export of LIM-kinase 1, mediated by two leucine-rich nuclear-export signals within the PDZ domain. *Biochem J* (1999) **3**:793-8.

And finally, the variants were differently phosphorylated/activated by upstream kinases as again demonstrated in Figure 6A. There are several possibilities of how mutating the LIMK PDZ domain can lead to changes in cofilin phosphorylation and yeast viability. A physical interaction between the LIMK PDZ and kinase domains is only one of them. Please discuss this accordingly.

>RESPONSE: We thank the reviewer for highlighting this important consideration which we did not adequately discuss in the initial submission. We agree that there are several possibilities for how changes in the PDZ domain may impact cofilin phosphorylation and yeast viability, and that the physical interaction is but one of these. Some of these other possibilities are altered protein expression or stability and changes in LIMK recognition as a substrate of upstream kinases or phosphatases. We assessed protein expression and stability in experiments shown in **Reviewer Figure R2** and find reduced expression levels for some mutants (e.g. L165A and R222A), but that these consistently show increased phosphorylation compared to wild-type. Altered recognition of LIMK by kinases and phosphatases would presumably be allosterically tethered to intramolecular conformational rearrangements which we are inclined to suggest may be associated with head-to-tail LIMK interactions, which we discuss in our response below associated with **Reviewer Figure R3**. We have updated our discussion on page 11 as shown below.

Page 11. "... Mutation of these residues caused the isolated PDZ domain to be insoluble when expressed in bacteria, similar to some of the surface residues (**Figure S8**), suggesting that proper folding of the PDZ domain is required for autoregulation. **It is important to note, however, that while our analyses provide a clear demonstration that the LIMK PDZ domain is important for changes in the activation loop phosphorylation of the LIM domain kinases and consequent changes in kinase activity, the work does not formally prove that this is mediated by a direct PDZ-kinase domain interaction. Although we consider this to be the likeliest possibility that results in changes in LIMK activation loop phosphorylation, other potential mechanisms include altered protein expression or stability, and changes to recognition of the LIMK as a substrate by upstream kinases and phosphatases. Notwithstanding these caveats, our studies demonstrate that a previously unidentified and completely conserved surface on the properly folded PDZ domain is required for normal autoregulation of the LIMKs.**"

iii Phosphorylation in T508 switches the LIMK1 catalytic domain from inactive to active. Is this also valid for the FL LIMK1? Could you please hypothesize on the interplay between T508 phosphorylation and PDZ autoregulation?

>RESPONSE: We thank the reviewer for raising these points. The central question here is whether activation loop phosphorylation drives the transition of the inactive full-length kinase to active full-length kinase, a mechanism that has long been assumed in the field, but we believe not adequately studied. Indeed, this is a key point in understanding the mechanisms of LIMK regulation, and for example, whether there are multiple linked conformational movements at play.

We have conducted new experiments (**Figure 7**) and introduced new discussion points in the text on pages 3, 8 and 10 to address this set of queries, which we expand upon in detail below.

Previous work has studied the impact of phosphorylation on LIMK activity. In a study by Ohashi and colleagues (citations listed below) the phosphorylation of full-length LIMK was monitored. The study found that Thr508 in the LIMK activation loop was phosphorylated by the upstream kinase ROCK, and that pre-treatment with ROCK increased LIMK activity. To make a phosphomimetic, Ohashi and colleagues introduced a double Glu (T508EE) in place of the activation loop residue T508. LIMK1 T508EE mutant showed higher activity compared to a LIMK1 without ROCK treatment and the constitutively inactive mutant T508V. Similar results were obtained for full length LIMK2 by Sumi and colleagues. The previous studies therefore suggest that activation loop phosphorylation and the corresponding phosphomimetic mutations increase kinase activity in the context of full-length protein. We have expanded our discussion of these previous results on page 3 as below:

Page 3. “.... Early studies suggested that the N-terminal domains play roles in regulation of catalytic activity. For example, truncated LIMK has elevated activity compared to full-length protein *in vitro* and in cultured cells, and the N-terminal non-catalytic region diminished catalytic activity of the isolated kinase domain in trans. **Similarly, mutation of the activation loop threonine to an unphosphorylatable residue results in suppression of activity, but phosphomimetic mutation increases activity. Yet, despite these findings the molecular basis for this suppression of activity remains unclear.**”

To address the reviewer’s query regarding interplay between activation loop phosphorylation and regulation by the PDZ domain, we asked if the PDZ domain could still autoregulate LIMK in the context of non-phosphorylatable or phosphomimetic mutations. We generated FL LIMK1 constructs in which Thr508 had been mutated either to valine (to prevent phosphorylation) or substituted with two Glu residues (to mimic phosphorylation). We further made compound E225A PDZ domain mutants in both backgrounds. In yeast growth assays, neither FL LIMK1-T508V nor FL LIMK1-T508V/E225A suppressed growth. In the context of the isolated catalytic domain, the T508V mutant did suppress growth, but not to the same extent as the WT catalytic domain (**Reviewer Figure R3**, shown below). These results are consistent with the T508V mutation reducing LIMK1 kinase activity both in the context of the catalytic domain and the FL protein as has been reported. However, they are ambiguous as to whether E225A increases LIMK1 activity in the absence of activation loop phosphorylation, since T508V/E225A may simply have insufficient activity to suppress growth. For the phosphomimetic mutants, we found that

expression FL LIMK1-T508EE did suppress growth, but to a lesser extent than did WT FL LIMK1. This result suggests that the T508EE mutation does not completely substitute for authentic phosphorylation, as is frequently the case with phosphomimetic kinase mutations. Importantly however, the combined T508EE/E225A reduced growth more severely than the T508EE mutant alone. Consistent with these observations, we found that the compound T508EE/E225A mutant had higher kinase activity measured in vitro than did FL LIMK1 T508EE. We have introduced these results in a new figure in the manuscript, **Figure 7**, shown below. We believe these results demonstrate that neither the PDZ domain nor the activation loop are solely responsible for full activation, and that multiple linked conformational movements are required to achieve full activity of the LIMKs. We have discussed these experiments on pages 8 and 10, as shown below.

Reviewer Figure R3. Yeast viability assay for LIMK1 T508V mutant. Serial dilutions of *cof1Δ* yeast expressing human cofilin and human LIMK1 with indicated mutants. Cofilin or control cofilin-S3A expressed in all cultures. LIMK1 full-length, catalytic domain and full-length mutated at E225A all include activation loop T508V mutation. The corresponding LIMK2 residue for T508 is T505, and for E225 is E206. Five-fold dilutions of yeast cultures were plated on solid media in the presence of glucose (-Gal) or galactose and raffinose (+Gal/Raff) to induce LIMK1 expression. Plates were grown at 30°C for 2 days (glucose plate) or 4 days (galactose plate).

was carried out using a non-parametric unpaired Mann-Whitney test. Two stars (*) indicates $p < 0.0286$. A total of 4 replicates were analyzed using GraphPad Prism.

Page 8. "...We observed higher activation loop phosphorylation in cell lysates from our yeast growth assays (**Figure 6**) as well as in purified protein used for our kinase assays (**Figure S9**). **To examine whether increased activation loop phosphorylation could account for elevated activity of PDZ domain mutants, we introduced point mutations into the activation loop that have previously been shown to mimic (T508EE; replacement of threonine-508 with two glutamic acids), phosphorylation-associated alterations in LIMK catalytic activity. In contrast, although we found that introduction of the phosphomimetic mutation T508EE into the activation loop of LIMK1 suppressed growth to a lesser extent than WT LIMK1 (presumably because it does not fully simulate phosphorylation), addition of E225A resulted in reduced viability, suggestive of increased catalytic activity (Figure 7a,b). Similarly, the compound mutant LIMK1 (E225A/T508EE) displayed stronger phosphorylation of cofilin *in vitro* than the activation loop phosphomimetic mutation alone (Figure 7c).**"

Page 10. "... Our studies strongly imply that the β A- β F- β D surface, and particularly a conserved glutamic acid, E206 (LIMK2) / E225 (LIMK1), are critical for autoregulation, **and that this regulation seems to be independent of activation loop phosphorylation.** Importantly, we found that surface mutations outside of this region and in the β B- α B cleft do not impact activity."

In summary, we believe that there is a two-step mechanism of activation for the LIM domain kinases; that the N-terminal region interacts with the kinase domain independently of activation loop phosphorylation. In schematic form, we believe **Reviewer Figure R4** below to be broadly correct, but have refrained from including this in the manuscript as it is probably too speculative. We have, rather, expanded our discussion of the regulation mechanism in comments on page 11, and shown below.

Reviewer Figure R4. Schematic illustrating potential steps of LIMK regulation. Left hand side: We believe our studies show that the fully autoinhibited LIMK is both unphosphorylated and has a potential direct interaction between the PDZ and kinase domains (although we have not formally demonstrated this last point). Right hand side: We believe our studies show that the fully active state is both activation loop phosphorylated and disrupted in the interaction of the PDZ and kinase domain. Top and bottom illustrate our finding that mutation of the PDZ domain can increase catalytic activity of either wild-type or T508EE LIMK.

Page 11. “Nonetheless, the changes in kinase activity that we observe suggest that disruption of the surface that potentially mediates autoregulatory interactions between the PDZ domain and the kinase domain allows LIMK to reach a more “open” conformation, suggesting **multiple independent, or partially independent, steps are required to fully activate the LIM domain kinases, including both disruption of N-terminal domain interactions with the kinase domain and activation loop phosphorylation by upstream activators.** Overall, our study clearly demonstrates that a previously unidentified surface on the PDZ domain plays a pivotal role in autoregulation of the LIM domain kinases.”

Ohashi, K., Nagata, K., Maekawa, M., Ishizaki, T., Narumiya, S. & Mizuno, K. Rho-associated kinase ROCK activates LIM-kinase 1 by phosphorylation at threonine 508 within the activation loop. *J Biol Chem* (2000) **275**, 3577-82

Sumi, T., Matsumoto, K. & Nakamura, T. Specific activation of LIM kinase 2 via phosphorylation of threonine 505 by ROCK, a Rho-dependent protein kinase. *J Biol Chem* (2001) **276**, 670-6

More specific comments:

Line 61. It may make sense to mention TESK1/2 here.

>RESPONSE: We thank the reviewer for this prompt. Indeed, based on the sequence similarities of the kinase domains, the LIMK family of proteins also includes the testis-specific kinase 1 and 2 (TESK1 and TESK2); the TESK kinase domains share approximately 50% sequence identity with the LIMKs. Importantly, however, with respect to this study, the non-catalytic domains of the TESKS differ significantly from the LIMKs, suggesting different mechanisms of regulation. We do not believe that our study of the LIMK PDZ domain to be relevant to understanding the details of the TESK regulation (albeit conceptually the role of inter-domain interactions in autoregulation may hold across the family to the TESKS). We have updated the text on page 3 to address this comment:

Page 3. "... Multiple Rho-effector protein kinases, including the ROCK, PAK and MRCK groups phosphorylate and activate the LIMKs^{4,6,7}. Importantly, the LIMKs (**and the TESKs which are related in their catalytic domains⁸**) appear unique in their ability to phosphorylate residue serine-3 of the actin depolymerizing factor, cofilin, which results in its inactivation...."

Line 75. Please explain the difference between the LIMK 'autoinhibited state' and the LIMK 'inactive state'.

>RESPONSE: We thank the reviewer for raising this important point. We were careful in the original manuscript to refer to a LIMK autoinhibited state rather than an inactive state. We believe this to be a semantically correct in the case of LIMK where the 'inactive state' would presumably be indistinguishable from the 'autoinhibited state' assuming the regulation is intra-molecular and that no other moieties are directly responsible for catalytic inhibition. In the original submission we did use the word 'inactive' once, in the legend of **Figure 5a**. Here we were referring to the kinase domain which has a 'catalytically inactive' mutant D460N that essentially removes the

ability of the enzyme to function, and we have chosen to retain this phrasing. We have shown this phrasing below in bold.

Legend for Figure 5. a) Full-length wild type (WT), kinase domain alone (CAT), and catalytically inactive kinase domain (CAT D460N) were used as positive and negative controls, respectively. PDZ domain mutants of conserved residues in FL LIMK1 are shown (equivalent residue in LIMK2 shown in parentheses).

Line 122. Please discuss why these boundaries were chosen. Was the LIMK2 AlphaFold model considered? Starting from D120 would have left the beta sheet intact. And overall, the LIMK LIM2 and PDZ domains seem to have merged into one single domain - this may also explain the unique sequence features of the LIMK PDZ domain.

>RESPONSE: We designed and generated the expression constructs for the PDZ domain prior to the release of AlphaFold in November 2020. Instead, we used a combination of conservation analysis and secondary prediction to design the domain boundaries. Based on our structure and on the structure of LIMK2 in the current version of AlphaFold (AF-P53671-F1-v4), D120 is located outside of the folded PDZ domain. For comparison, our built structure begins at residue Q145, slightly N-terminal to the first β -strand of the PDZ domain. Our structure of the LIMK2 PDZ domain has reasonable similarities to canonical PDZ domains, comprising the expected canonical six β -strands and the canonical α B helix (as discussed on page 5). We consider our boundaries to be appropriate for the PDZ domain.

On the topic of the LIM2 and PDZ domains potentially merging into a single bi-lobed domain, this is a possibility, and we discuss this in more detail in our response to the query regarding Line 282 below.

Line 191. Please discuss whether the differences in yeast viability between CAT and WT can rely on differences in the T508 phosphorylation grade.

>RESPONSE: We thank the reviewer for this query as it aims at deciphering the importance of activation loop phosphorylation versus autoregulation in the activity of LIMK and consequently yeast viability. In response, we have expanded our study to assess the impact of phosphorylation

mimetic T508EE on yeast viability and have also assessed the impact of T508V mutation, a phosphorylation-resistant mutation that maintains a beta-branched sidechain. We discuss these results in our response to query (iii) above. In summary, we found that even in the context of the phosphorylation resistant or phosphomimetic mutations, growth suppression is more severe for the catalytic domain alone than it is for FL LIMK1 (shown in **Figs 7** and **S10** of the revised manuscript). Our results suggest that while essential to achieve full activation of the kinase, phosphorylation of the activation loop cannot bypass the autoregulatory effects of the non-catalytic domains. We have expanded our discussion of these previous results in the text and in our response to query (iii) above.

Line 205. In Figure S5, all LIMK1 variants appear to have similar expression levels. Is this in contrast to Figure 6A?

>RESPONSE: We have answered this query in our response to query (ii) above where we discuss the differences in expression levels of the original **Figures 6A and S5**.

And surprisingly, the kinase activities of all variants appear to be similar, with no sign of autoinhibition via the PDZ domain.

>RESPONSE: The reviewer bases this comment on observations that the level of cofilin pSer3 in yeast is similar for all LIMK constructs expressed, including PDZ mutants (**Figure S7**). We believe this to be the case because growth inhibition likely requires phosphorylation of the large majority of cofilin molecules. In this case, small differences in absolute level of phospho-cofilin, which correspond to comparatively larger differences in the level of *non*-phosphorylated cofilin, may yield different growth rates. Such differences would be difficult to determine by immunoblotting for phospho-cofilin levels. To address this issue, we further examined the phosphorylation status of cofilin using Phos-tag gel analysis, in which the phosphorylated and non-phosphorylated species resolve from each other. By this analysis, we can make a more accurate ratiometric measurement of the stoichiometry of phosphorylation. Over an induction time course, we found partial phosphorylation at 2 hours post-LIMK induction, making it an optimal time to observe differences in cofilin phosphorylation among mutants. We found that all of our LIMK1 PDZ domain mutants, as well as the LIMK1 catalytic domain, induced a significantly higher degree of cofilin

phosphorylation than did WT FL LIMK1. We have added new panels to **Figure 4** and have introduced new text on page 7 to reflect these new experiments.

Figure 4: PDZ domain mutants suppress yeast growth and increase cofilin phosphorylation. a) Serial dilutions of *cof1Δ* yeast expressing human cofilin and the indicated human LIMK1 mutants. Controls of human LIMK1 constructs, full-length (FL), kinase domain (CAT), unphosphorylatable cofilin S3A (S3A) and empty vector (EV). Mutants of full-length LIMK1: E225A, D221A, R222A, L165A, Q251A. Corresponding LIMK2 residue is shown in parentheses. Five-fold dilutions of yeast cultures were plated on solid media in the presence of glucose (-Gal) or galactose and raffinose (+Gal/Raff) to induce LIMK1 expression. Plates were grown at 30°C

for 2 days (glucose plate) or 4 days (galactose plate). Representative of 3 independent experiments. **b)** Mutants assessed are shown on the cartoon and surface representations of the conservation map of the LIMK2 PDZ domain. Residues shown and equivalent human LIMK1 residue numbers: L152 (L165 in LIMK1), Q232 (Q251 in LIMK1), D202 (D221 in LIMK1), R203 (R222 in LIMK1) and E206 (E225 in LIMK1). **c)** Immunoblots of lysates corresponding to yeast plated in (a). Cofilin species are separated based on phosphorylation state by Phos-tag SDS-PAGE. Kss1 serves as a loading control. Images are representative of N=4. **d)** Quantification of immunoblots measuring the percentage of total unphosphorylated cofilin. Statistical analysis was carried out using a non-parametric unpaired Mann-Whitney test. One star (*) indicates $p < 0.05$. A total of 4 replicates were analyzed using GraphPad Prism.

Page 7. “To assess whether these alterations in yeast growth were indeed due to changes in LIMK catalytic activity, we **examined the level of cofilin phosphorylation following LIMK induction by Phos-tag SDS-PAGE. We observed that PDZ domain mutations increased the proportion of phosphorylated cofilin in yeast (Figure 4c,d). As this analysis suggested increased kinase activity, we directly assessed the impact of LIMK mutations on phosphorylation of cofilin in vitro.** We purified FL WT LIMK1 and the”

Page 15. **“Phos-tag SDS-PAGE analysis**

To assess the impact of LIMK1 expression cofilin phosphorylation, yeast were grown overnight at 30°C in SC-His-Leu containing 2% glucose. The following day, cultures were diluted into SC-His-Leu containing 2% raffinose and split into two groups. We added 1% galactose to one group to induce LIMK1 expression, and we added 1% glucose to the other group to repress LIMK1 expression. After 2 h, cells were harvested and lysed by TCA extraction as described above. Protein concentrations were determined by BCA assay, and 3 µg of each was fractionated using Phos-tag gels (SuperSep Phos-Tag 12.5% Cat #195-17991, and 7.5% Cat #192-18001, FUJIFILM Wako Chemicals). After fractionation, gels were incubated twice for 15 minutes in transfer buffer containing 10 mM EDTA to remove zinc ions before transfer to PVDF (Sigma, IPFL85R) membrane. Membranes were blocked and probed as described above with anti-Penta-His (QIAGEN, #34650) as the primary antibody.”

Line 213. Is it the low activity of non-phospho LIMK1 that is assessed here? Or is it the activity of pT508 LIMK1? The variant T508A would have been helpful. Please indicate whether or not the kinase activities correlate with the T508 phosphogrades.

>RESPONSE: The reviewer raises an interesting question regarding whether the low activity of WT FL LIMK1 is reflective of its low Thr508 phosphorylation in comparison to the PDZ domain mutants. We have addressed this comment in our response to query (iii) by conducting yeast growth assays for both T508V and T508EE mutants, and kinase activity assays for T508EE. We

have additionally introduced the PDZ mutant E225A into these assays. Our results clearly show that E225A elevates both kinase activity and suppresses yeast growth in the context of T508EE. In addition, the relatively small increase in activation loop phosphorylation conferred by E225A mutation would seem unlikely to account for the comparatively larger increase seen in kinase activity. Overall, we conclude that although kinase activity does correlate with the level of Thr508 phosphorylation, we believe there to be a two-step mechanism of activation, as shown in **Reviewer Figure R3**.

Line 224. Is this autophosphorylation? Also, the D460N variant is phosphorylated in Figure S7. Is it rather yeast kinases phosphorylating LIMK1?

>RESPONSE: We thank the reviewer for pointing out this error on line 224 where we previously stated “We found that in keeping with coordinated autophosphorylation and intramolecular interactions, activation loop phosphorylation was consistently elevated for point mutations that increased kinase activity.” In prior reports (e.g. Ohashi 2000 and Amano 2001, cited below) LIMK autophosphorylation does not occur on the activation loop, which is phosphorylated by distinct upstream kinases, and indeed the D460N mutant is phosphorylated by upstream kinases as the reviewer points out. We have modified the sentence on page 8 to remove the reference to autophosphorylation, as below.

Page 8. “We finally assessed the role of the PDZ domain in the regulation of LIMK activation loop phosphorylation. The steps of regulation for these kinases are not resolved, and it is still unclear how autoregulation and activation loop phosphorylation coordinate to regulate activity. Therefore, we wondered if introduction of these point mutations could impact the phosphorylation of the LIMK activation loop. **We found that in keeping with coordinated intramolecular interactions, activation loop phosphorylation was consistently elevated for point mutations that increased kinase activity.** We observe higher activation loop phosphorylation in both our yeast growth assays (**Figure 6**), and in our purified protein used for our kinase assays (**Figure S7**).....”

Ohashi, K., Nagata, K., Maekawa, M., Ishizaki, T., Narumiya, S. & Mizuno, K. Rho-associated kinase ROCK activates LIM-kinase 1 by phosphorylation at threonine 508 within the activation loop. *J Biol Chem* **275**, 3577-82 (2000).

Amano, T., Tanabe, K., Eto, T., Narumiya, S. & Mizuno, K. LIM-kinase 2 induces formation of stress fibres, focal adhesions and membrane blebs, dependent on its activation by Rho-associated kinase-catalysed phosphorylation at threonine-505. *Biochem J* **354**, 149-59 (2001).

Line 282. Another conclusion from the AlphaFold model is the high-confidence placement

of the LIM2 domain close to the conserved PDZ surface patch. Furthermore, in reference 19, the N-terminal LIM domains (and not the PDZ domain) are suggested to be the key regulators of LIMK kinase activity. How can this be followed up in the future?

>RESPONSE: Previous studies have indeed found that the LIM domains are important for regulation (see listed below: Hiraoka et al., 1996; Edwards, et al., 1999; Nagata et al., 1999; Arber et al., 1998). These studies relied heavily on mutation of the zinc binding sites within the LIM domains, which disrupts the folding of these dual zinc finger domains. Based on the AlphaFold models, which as the reviewer states, are high-confidence, we do not believe that these previous findings are inconsistent with our study. When our PDZ domain structure is superposed onto the LIM2-PDZ AlphaFold prediction, we find that the location of the mutated residues and particularly of E225, is on a contiguous conserved surface. This potentially suggests that the LIM2 domain may be contributing part of an interaction surface in addition to the PDZ domain, and we have commented on this finding in the discussion on page 11.

Page 11. "This work provides a first molecular level insight into the molecular surfaces important for autoregulation of the LIM domain kinases. Based on superposition of over 40 AlphaFold models of full-length LIMK1 and LIMK2 in different species, we found that the β A- β F- β D surface is almost completely surface exposed, with a small portion of the surface consistently found to interact with the adjacent LIM2 domain (residue L152 and residues of β A which makes an anti-parallel β -sheet interaction with the LIM2 domain). In these models, residue E206 (LIMK2) / E225 (LIMK1) is always surface exposed further supporting our finding that the β A- β F- β D surface has the potential **to regulate** the kinase, **and also allows for an extended surface consistent with previous literature suggesting a role for the LIM domains in autoregulation.** The changes in....."

Hiraoka, J., Okano, I., Higuchi, O., Yang, N. & Mizuno, K. Self-association of LIM-kinase 1 mediated by the interaction between an N-terminal LIM domain and a C-terminal kinase domain. *FEBS Lett* **399**, 117-21 (1996).

Edwards, D.C. & Gill, G.N. Structural features of LIM kinase that control effects on the actin cytoskeleton. *J Biol Chem* **274**, 11352-61 (1999).

Nagata, K., Ohashi, K., Yang, N. & Mizuno, K. The N-terminal LIM domain negatively regulates the kinase activity of LIM-kinase 1. *Biochem J* **343 Pt 1**, 99-105 (1999). PMC1220529

Arber, S., Barbayannis, F.A., Hanser, H., Schneider, C., Stanyon, C.A., Bernard, O. & Caroni, P. Regulation of actin dynamics through phosphorylation of cofilin by LIM-kinase. *Nature* **393**, 805-9 (1998).

Line 305. Several statements in this paragraph strongly suggest that the LIMK PDZ and kinase domains interacted physically. Please consider modifying the wording or provide additional experimental evidence for this claim.

>RESPONSE: We have edited the paragraph to remove these statements. The paragraph is pasted in our response to "Line 282".

Reviewer #2 (Remarks to the Author):

In their manuscript ‘Autoregulation of the LIM kinases by their PDZ domain’, the crystal structure of human LIMK2 PDZ domain was determined, with a shallow $\beta\text{B}-\alpha\text{B}$ binding groove, which may explain why the LIMK PDZ domains, unlike canonical PDZ domains, cannot interact with the carboxy-terminal peptides. Based on the sequence conservation and electrostatic potential analysis, the authors defined a hydrophobic patch, the $\beta\text{A}-\beta\text{F}-\beta\text{D}$ surface, conserved in all LIMK PDZ domains. Furthermore, they used human LIMK1 proteins ectopically expressed in yeast to examine the importance of this $\beta\text{A}-\beta\text{F}-\beta\text{D}$ surface on LIMK1 activity. This study provides insights into the autoregulation mechanism of the LIM kinases by a previously unidentified surface on the PDZ domain. However, revision of the manuscript to address the concerns below will make it more relevant to the broad readership of Nature Communications.

>RESPONSE: We thank the reviewer for their thoughtful and careful comments on our manuscript. Below we have addressed each of their comments and queries.

1. PDZ domains generally recognize the C-terminal residues of binding partners, while the LIMK PDZ domain likely does not interact with C-terminal peptides. The authors mentioned that potential binding partners of the LIMK PDZ domain have not been identified (line 91). The N-terminal LIM domains of LIMK were reported to interact with the C-terminal kinase domain and negatively regulate the kinase activity (ref. 18 and 19). Would the PDZ domain also interact with the kinase domain (and/or the LIM domains)? It would be helpful to carry out interaction analysis among the LIMK PDZ, LIM1/2 and kinase domains, as well as trans-inhibition assay of the PDZ domain and the N-terminal fragment containing both LIM and PDZ domains (WT and PDZ mutations) on the activity of kinase domain.

>RESPONSE: We thank the reviewer for this thoughtful comment and query. The reviewer requests two orthogonal assessments of interaction between the PDZ and LIM-PDZ domains and the kinase domain. In response to the query on trans-inhibition, we have assessed the ability of the PDZ and of the LIM2-PDZ regions to suppress the kinase activity of LIMK1 catalytic domain toward cofilin. Unsurprisingly for a potential intra-molecular autoinhibitory interaction, we find that inhibition by either the PDZ or the LIM2-PDZ domain is weak, with suppression of kinase activity visible at around 160 μM for both. Interestingly, we find that titration of the LIM2-PDZ domain is

better at suppressing activity, but still weak with ~50% inhibition at 300 μM (Reviewer Figure R5). We interpret this in the context of our AlphaFold analysis and response to reviewer 1 (line 282 comment), to suggest that the LIM2 domain may extend the interaction surface. We have discussed this in the text on page 11, as below.

Reviewer Figure R5. Kinase activity titrations. As expected, trans-inhibition of catalytic activity is weak. Nonetheless, inhibition by both PDZ and PDZ-LIM2 constructs is moderated by introduction of E225A mutation. **a)** Titration of LIMK1 catalytic domain with 160 M of LIMK1 PDZ domain or LIMK1 PDZ domain E225A. Top shows quantification of cofilin phosphorylation and bottom shows autoradiography and Coomassie stain. **b)** Titration of LIMK1 catalytic domain with LIMK1 LIM2-PDZ region. Top shows quantification of cofilin phosphorylation and bottom shows autoradiography and Coomassie stain.

Page 11. “This work provides a first molecular level insight into the molecular surfaces important for autoregulation of the LIM domain kinases. Based on superposition of over 40 AlphaFold models of full-length LIMK1 and LIMK2 in different species, we found that the βA - βF - βD surface is almost completely surface exposed, with a small portion of the surface consistently found to interact with the adjacent LIM2 domain (residue L152 and residues of βA which makes an anti-parallel β -sheet interaction with the LIM2 domain). In these models, residue E206 (LIMK2) / E225 (LIMK1) is always surface exposed further supporting our finding that the βA - βF - βD surface has the potential **to regulate** the kinase, **and also allows for an extended surface consistent with**

previous literature suggesting a role for the LIM domains in autoregulation. The changes in kinase activity that ...”

In response to the query regarding analysis of direct interactions, unfortunately, our analysis of direct interaction between the PDZ or LIM2-PDZ domains with the catalytic domain has been hampered by low expression yields. We would ideally like to perform isothermal titration calorimetry, or surface plasmon resonance, for these samples – gold-standard experiments which require preparations of large amounts of pure, homogeneously phosphorylated, and stable protein. However, we are not able to purify sufficient quantities of both PDZ/LIM2-PDZ and kinase domain for either protein to perform our desired biophysical analyses.

We believe that one of the key motivating factors behind Reviewer 2’s query may have been whether there is a direct interaction between the PDZ or LIM2-PDZ region and the catalytic domain. While we believe this to be so, we acknowledge that we have not formally proven this to be the case. We have therefore edited our text to remove references to direct interactions between the PDZ and kinase domain, and to discuss structural and biophysical assessment of direct interactions as a future goal of studies of the LIMKs (page 11). We hope the reviewer agrees this to be sufficient to describe our experimental observations.

Page 11. “...Nonetheless, the changes in kinase activity that we observe suggest that disruption of the surface that potentially mediates autoregulatory interactions between the PDZ domain and the kinase domain allows LIMK to reach a more “open” conformation, suggesting multiple independent, or partially independent, steps are required to fully activate the LIM domain kinases, including both disruption of N-terminal domain interactions with the kinase domain and activation loop phosphorylation by upstream activators. **A detailed biophysical exploration of these potential direct interactions is therefore warranted.** Overall, our study clearly demonstrates that a previously unidentified surface on the PDZ domain plays a pivotal role in autoregulation of the LIM domain kinases.”

2. There are more than 250 PDZ domains in over 150 human proteins, which could bind short linear peptides or non-peptide ligands. The authors found the LIMK PDZ domain adopting a canonical PDZ fold with an unusual non-canonical mechanism for peptide/partner recognition. Please introduce the structural features/differences of canonical and non-canonical PDZ domains, as well as the different binding modes (PDZ domains can mediate protein-protein interactions through alternate modes or alternative binding surfaces).

>RESPONSE: This is an excellent suggestion. We have now included a more in-depth discussion of PDZ domains on pages 4, 10, and in the new supplemental figure, **Figure S1**. We thank the reviewer for providing us with an opportunity to better position the LIMK PDZ domain in the context of the known binding modes and protein-protein interactions of canonical and non-canonical PDZ domains. We believe this has improved the accessibility of our manuscript.

Page 4. “The PDZ domain was named after its early identification in three proteins (postsynaptic density 95, PSD-95; discs large, Dlg; zonula occludens-1, ZO-1)¹⁰⁻¹⁴, and more than 250 examples have been found in over 150 human proteins¹⁵⁻¹⁷. Generally, these non-catalytic domains are thought to mediate protein-protein interactions, typically by specific recognition of linear peptide motifs in the C-terminal tails of protein binding partners¹⁸⁻²³. The LIMK PDZ domain is unusual however in that it does not interact tightly with carboxy-terminal peptides^{17,24}. PDZ domains can also mediate protein interactions through alternate modes, including interactions of the canonical binding site with internal peptide motifs, or use of alternative binding surfaces²⁵⁻³². **Many of the differences between canonical and non-canonical PDZ domains focus on a central binding site between two structural features of the domain, an α -helix and a β -strand (Figure S1a) and the abilities of non-canonical PDZ domains to bypass, modify or control these features (Figure S1b-e).** There is, therefore, a possibility that the LIMK PDZ domain might similarly use alternative binding surfaces for intermolecular protein-protein interactions, however interactions with potential binding partners have not been identified³³⁻³⁷. Early studies suggested that the PDZ domain may impact LIMK autoregulation^{3,5}, but it is still unclear whether this occurs through canonical interactions with the peptide binding cleft or through other binding surfaces.”

Page 10. “Our crystal structure **reveals new structural insights into the well-studied PDZ** fold. Comparison of the LIMK2 PDZ domain to other human PDZ domains revealed three unusual features suggestive of functional relevance. First, we observed that the canonical peptide binding cleft between the β B strand and α B helix is particularly shallow, and that the orientation of the α B- β F loop encroaches on the binding groove. While it is not necessarily unusual to observe a shallow cleft in PDZ domains (for example PDZ7 of GRIP³²) this feature provides a rationale for why the LIM domain kinase PDZ domains have so far not been found to interact with C-terminal peptides with biological range affinities in PDZ interaction screening studies^{17,24}. Second, we found that the second position of the ‘ χ - Φ -G- Φ ’ motif was unique in the entire PDZ fold – a hydrophobic core-facing arginine residue (Arg163 in LIMK2, and Arg176 in LIMK1). The hydrogen-bonding interactions of this stringently conserved arginine caps the α B helix, coordinates the α B- β F loop, and seems to provide a rigid base for the C-terminus of the α B helix. Third, we found that the α A helix is replaced by two 3_{10} helices. This combination of unusual features for the LIMK PDZ domain make it difficult to place into the previously assigned PDZ classes (classes I, II, III or IV^{27,38-41}) (**Figure S1a**). These features do, however, tempt conjecture that this domain could engage in bi-directional allostery. Previous studies (e.g. the interaction between Cdc42 and Par6^{42,43}) have found that binding partner interactions, often with helix α A, can increase carboxylate peptide binding affinity and vice versa (**Figure S1b-e**). It is therefore interesting to speculate that the LIMK PDZ domain may be primed for carboxylate peptide binding, but require allosteric-induced conformational movements to reveal the high-affinity binding site. Further studies will be needed to probe this more fully.”

Figure S1. Canonical and non-canonical PDZ domains. a) Schematic cartoon illustrating binding of a canonical PDZ domain to a C-terminal peptide. b-e) Non-canonical PDZ domain interactions illustrated in cartoon format, including peptide binding to internal and extension motifs (b) regulation by phosphorylation (c), noncanonical interaction surfaces (d), and allosterically induced conformational changes that alter binding partner interactions (e).

The structure of the human LIMK2 PDZ domain contains 8 copies per asymmetric unit. What is the oligomeric state of the human LIMK2 PDZ domain in solution?

>RESPONSE: We thank the reviewer for this query. This is an important question regarding the ability of the PDZ domain of LIM kinases to drive dimerization or oligomerization, and whether the crystal structure provides hints towards potential mechanisms if this were to be the case. We

acknowledge that the potential of the PDZ domain to drive homotypic interactions may provide an orthogonal angle by which to query to regulation of LIMKs. Nonetheless, we do not believe that homotypic interactions are critical. We base this assessment on a couple of experimental findings. First, if a homotypic PDZ domain interaction were critical for normal LIMK function we would expect the residues mediating the interaction to be conserved, and for the interaction to recapitulate in the crystal structure, however, the crystal packing is not mediated by conserved residues suggesting a lack of homotypic driven interaction (**Reviewer Figure R6**). Second, when we assessed the oligomerization state of the LIM2-PDZ region (expected MW 21.4 kDa) using in-line size exclusion chromatography-small-angle X-ray scattering and observe a monomeric species by Porod volume (MW 18.9kDa) and by Volume of Correlation (MW 20.6 kDa). Below we show the SAXS data in **Reviewer Table R1**) and **Reviewer Figure R7**). We believe our analyses demonstrate that the LIMK PDZ domain does not drive oligomerization of LIMKs, and have stated this on page 5:

Page 5. “We expressed, purified, and crystallized the human LIMK2 PDZ domain (residues 145-236), **which is monomeric in solution**, and determined its structure to 2.0 Å resolution (**Figure 2a, Table 1**).”

Reviewer Figure R6. Crystal packing of LIMK2 PDZ domain colored by conservation. The crystal packing of LIMK2 PDZ when colored by surface conservation illustrates that the majority of the highly conserved residues are solvent exposed, a finding compatible with a monomeric PDZ domain.

Reviewer Figure R7. Small-angle X-ray Scattering for LIMK2 LIM2-PDZ. In-line SEC-SAXS was performed at LiX beamline at NSLS-II. **a)** SAXS scattering profile averaged across the protein elution peak. **b)** Guinier analysis (top) and the plotted residuals between the data and fit (bottom). **c)** Kratky analysis. **d)** Porod-Debye analysis, flexibility is suggested as the curve fails to plateau. **e)** Pair distribution functions. **f)** DENSS reconstruction showing similarity of SAXS data with monomeric LIMK2 LIM2-PDZ model generated using AlphaFold.

Reviewer Table R1. SAXS data collection and analysis parameters for LIMK2 LIM2-PDZ.

Organism	Human
Source	E. coli
Sequence of construct	GSPKDYWGKFGFEGCHGCSLLMTGPFMVAGEFKYHPECFACMSCKVIIEDGD AYALVQHATLYCGKCHNEVVLAPMFERLSTESVQEQLPYSVTLISMPATTEG RRGFSVSVESASSNYATTVQVKEVNRMHISPNNRNAIHPGDRILEINGTPVR TLRVEEVEDAISQTSQTLQLLIEHDPVSQRDLQLRLE
Extinction coefficient ϵ ($M^{-1} cm^{-1}$)	14440
MW (kDa)	21.4
Loading concentration ($mg mL^{-1}$)	11
Injection volume (μL)	60
Flow Rate ($ml min^{-1}$)	0.35
Solvent composition	150mM NaCl, 20mM Tris pH 8, 1mM DTT
SAXS Data Collection Parameters	
Instrument	BNL, NSLS-II, LiX Beamline, sector 16-ID
Wavelength (\AA)	0.819
Camera length (m)	3.686
Beam size	150 (h) x 25 (v) focused at the detector
q-measurement range (\AA^{-1})	$0.006 < q < 3.0 \text{\AA}^{-1}$
Absolute scaling method	Glassy Carbon, NIST SRM 3600
Basis for normalization to constant counts	To transmitted intensity by beam-stop counter
Method for monitoring radiation damage	Automated frame-by-frame comparison of relevant regions using CORMAP (Franke et al., 2015)[91] implemented in BioXTAS RAW
Sample configuration	SEC-MALS-DLS-RI-SAXS. Size separation used a Superdex 200 Increase 5/150 GL column (Wyatt Technology) and a 1260 Infinity II HPLC (Agilent Technologies). UV data was measured in the Agilent
Exposure time (s)	0.5
Exposure period (s)	2
Sample temperature ($^{\circ}C$)	22
Software employed for SAXS data reduction	
SAXS data reduction	Radial averaging; frame comparison, averaging, and subtraction done using BioXTAS RAW 2.0.3 (Hopkins et al., 2017)[90]
Basic analysis: Guinier, M.W., P(R)	Guinier fit and M.W. using BioXTAS RAW, P(r) function using GNOM (Svergun, 1992)[92]. RAW uses MoW and Vc M.W. methods (Rambo & Tainer, 2013; Piiadov et al., 2018)[93, 94]
e from sequence	ProtParam Tool - ExPASy
Electron density	Performed in RAW v2.1.3 according to (Grant 2018)
Molecular graphics	CCP4mg
Structural parameters	
Guinier Analysis	
$I(0)$ (cm^{-1})	0.083
R_g (\AA)	19.8
q-range (\AA^{-1})	0.022 to 0.068
P(R) Analysis	
R_g (\AA)	20.25
D_{max} (\AA)	80
q-range (\AA^{-1})	0.005 - 0.04
Porod Volume (V_p) MW (kDa)	18.8
Volume of Correlation (V_c) MW (kDa)	20.6
DENSS Reconstructions	
Chi squared value	1.55
Model R_g (\AA)	19.38
Model resolution (\AA)	24.35 +/- 4.55

Does the crystal packing give potential binding/recognition mode(s) of LIMK PDZ domain and partner?

>RESPONSE: We do not believe that the crystal packing of the LIMK PDZ domain provides further insight into potential binding and or recognition modes of interaction. Crystal packing does not occur in the proximity of the canonical binding site, and similarly does not occur at the β A- β F- β D surface. We have briefly discussed this on page 6 as shown below.

Page 6. “Canonical PDZ domains utilize the β B- α B groove to bind partner peptides and coordinate terminal carboxylate groups through backbone amide interactions of the central Φ -G residues of the ‘x- Φ -G- Φ ’ motif. The inward orientation of Arg163 to cap helix α B seems to be key for orientations of the β A- β B and α B- β F loops, **and crystal packing does not seem to impact these orientations**. In addition, an inward orientation of helix α B and placement of Arg163’s C β atom to encroach on the expected carboxylate binding site provides a potential explanation for why the LIMK PDZ domains do not interact with carboxy-terminal peptides with measured affinities in a biological range^{17,24}.”

3. The ‘G-L-G-F’ or ‘X- Φ -G- Φ ’ motif was first mentioned without a detailed description (line 117).

>RESPONSE: We thank the reviewer for noticing this omission. We have now provided a detailed description on page 5 as shown below.

Page 5. “To explore the role of the LIMK PDZ domain, we generated a sequence alignment of the PDZ domain from 421 LIMK orthologs across animal species with a set of canonical PDZ domains from other proteins. We found high conservation of the LIMK PDZ domain between human LIMK1 (residues 159-258) and human LIMK2 (residues 147-239), which are 47% identical and 81% similar. This high conservation is maintained across species, with the human LIMK1 PDZ being 36% identical to that of the *D. melanogaster* ortholog for example. There was lower sequence similarity to canonical PDZ domains, (21% identical to PSD95). Interestingly, one of the defining features of canonical PDZ domains **was divergent in all LIMK orthologs; this motif is termed the ‘G-L-G-F’ motif (after a sequence in the PSD-95 protein) or more generally termed the ‘x- Φ -G- Φ ’ motif, where x represents any, and Φ represents hydrophobic amino acid**^{14,44,45} (Figure 1B, Figure S1). To investigate....”

Canonical PDZ domains utilize the β B- α B groove to bind partner peptides and terminal carboxylate groups, while the inward-facing arginine residue of the human LIMK2 PDZ domain encroaches on the expected carboxylate binding site resulting in a shallow binding

groove. Please provide a structural comparison of the human LIMK2 PDZ domain and a canonical PDZ domain bound with a peptide to illustrate the significance of this Arg residue.

>RESPONSE: We thank the reviewer for this request. The LIMK PDZ domain is indeed a structurally unique PDZ domain, and this seems to be driven by the arginine residue at the GLGF motif's second position (Arg 176 LIMK1, Arg 162 LIMK2). In our original submission, we included a comparison between PDZ domains illustrating the main chain differences between LIMK2 PDZ, and three canonical PDZ domains (**Figure 2D**), but we realize now that this analysis may not have adequately described the unusual nature of the LIMK β B- α B groove. As the reviewer suggests, to better illustrate the unique structural properties of the LIMK PDZ, we have generated a new figure superposing LIMK2 PDZ domain referred to on pages 6-7. PDZ domains in complex with their respective peptide binding partners (**Figure S5**). The introduction of the arginine residue seems to generate a backbone conformational movement which encroaches on the canonical peptide binding location and generates a steric clash. We have added a new supplemental figure as shown below and highlight the relevant passage from the text on pages 6-7.

Supplemental Figure S5. Canonical vs LIMK PDZ binding. Comparison of the α A- β F loop orientation of LIMK2 PDZ crystal structure (orange) and the most similar PDZ domains structures as designated by Dali search⁴⁶; membrane-associated guanylate kinase with inverted domain structure protein 1 (MAGI-1) PDZ2 bound to RSK1 peptide, PDB ID: 5N7D⁴⁷ (dark grey), The third PDZ domain from the synaptic protein PSD-95in complex with a C-terminal peptide derived from CRIPT, PDB ID: 5HEY⁴⁸ (grey), PDZ domain from Human microtubule-associated serine/threonine-protein kinase 1(MAST1) in complex with a class I C-terminal peptide sequence, PDB ID: 3PS4 (light grey). Images generated using CCP4mg⁴⁹.

Page 6-7. “Alignment over all human PDZ domains indicated that the LIMKs are the only PDZ domains harboring an arginine residue in the second position of the χ - Φ -G- Φ motif. The residue at this position is normally oriented toward the hydrophobic core of the domain. Unusually for a charged residue, we found Arg163 in a similar orientation. To balance the charge of the guanidino group, Arg163 engages in extensive hydrogen bonding. It caps helix α B, hydrogen bonds to the carboxyl oxygens of residues Ala223, Ile224 and Gln226, and also hydrogen bonds to the carboxyl oxygen of Gln229 within the α B- β F loop (**Figure 2b,c**). This arrangement seems to provide a rigid anchor for the C-terminus of the α B helix. A consequence of this inward-facing

arginine residue is that it helps create a somewhat shallow binding groove between the β B strand and α B helix (**Figures 2d, S5**). Canonical PDZ domains utilize the β B- α B groove to bind partner peptides and coordinate terminal carboxylate groups through backbone amide interactions of the central Φ -G residues of the ' χ - Φ -G- Φ ' motif. The inward orientation of Arg163 to cap helix α B seems to be key for orientations of the β A- β B and α B- β F loops, and crystal packing does not seem to impact these orientations. In addition, an inward orientation of helix α B and placement of Arg163's C β atom to encroach on the expected carboxylate binding site provides a potential explanation for why the LIMK PDZ domains do not interact with carboxy-terminal peptides with measured affinities in a biological range^{17,24}.

4. The newly-identified β A- β F- β D surface is conserved among all LIMK orthologs. The authors generated point mutations of this surface on human LIMK1, instead of human LIMK2, to examine its importance?

>RESPONSE: This is a similar query to item (i) raised by reviewer 1. We refer the reviewer to that response and summarize that we began by focusing on LIMK1 exclusively, which drove the development of our yeast assays, were not able to crystallize LIMK1 PDZ but were able to crystallize LIMK2 PDZ, and based on sequence conservation-based experimental design believe we have revealed a conserved mechanism of LIMK regulation. In response to this query and the similar query of Reviewer 1, we attempted to assess LIMK2 in our yeast growth assay and cofilin phosphorylation experiments, but find that LIMK2 is not as active in yeast as LIMK1 (**Reviewer Figure R1**), preventing quantitative analyses of effects of these mutations in LIMK2 in this system.

The PDZ domains of human LIMK1 and LIMK2 are highly conserved. How are two full-length proteins, sequence conserved and function redundantly?

>RESPONSE: We thank the reviewer for this query and have made edits to the text to emphasize both the similarities and differences of LIMK1 and LIMK2. The *LIMK1* and *LIMK2* genes are located on chromosomes 7q11.23 and 22q12.2, respectively. Their expression patterns differ, with LIMK1 showing higher expression in the brain, kidney, lung, stomach and testis, while LIMK2 has a broader expression profile observed in both adult and embryonic tissue. While the surface conservation of the PDZ domain discussed in this study and the substrate recognition surface of the kinase domain discussed in our prior work (Hamill et al., 2016) both show complete conservation of important regions of these proteins, we do not believe the surfaces of these proteins to be completely conserved across the whole protein raising the possibility for other

protein actors to differentially target LIMKs for interaction, localization and/or regulation. This is illustrated, for example, in the differing phosphorylation profiles of LIMK1 and LIMK2 (<https://www.phosphosite.org/proteinAction?id=616> and <https://www.phosphosite.org/proteinAction?id=828>). We have expanded our discussion on this topic in the text on page 3 as shown below.

Page 3. “LIMKs are found in most animal species but are absent from fungi and plants, **in humans their expression profiles differ, with LIMK1 showing higher expression in the brain, kidney, lung, stomach and testis, and LIMK2 with broader expression in both adult and embryonic tissue.** LIMKs across species have a common architecture, with two N-terminal tandem-zinc finger LIM domains followed by a PDZ domain, a predicted unstructured region enriched in serine, proline and glycine residues, and a C-terminal kinase domain (**Figure 1A**). Like many other kinases, activation of these multi-domain enzymes is associated with phosphorylation...”

The β A- β F- β D surface is hydrophobic (line 169 and Figure 3B). Why the point mutations used (L165, E225, D221, R222 and Q251) involve only one hydrophobic residue?

>RESPONSE: On our analysis of the structure, we found that there was an extended conserved surface, the β A- β F- β D surface and to assess its importance we introduced point mutations in residues that are highly conserved through evolution across 421 sequences of LIMK1 and LIMK2. We based our analysis on the sequence conservation.

In line 204, human LIMK1 E225A mutations resulted in a complete loss of yeast growth, suggesting that the activity of this mutant LIMK1 comparable to the uninhibited kinase domain alone. As shown in Figure S5, the phosphorylation levels of p-Cofilin in yeast expressing LIMK1 WT and PDZ mutants are almost the same. The catalytic domain and LIMK1-WT were expected to have a dramatic difference. Moreover, the result of in vitro kinase activity assay showed that the E225A mutant of LIMK1 FL increased ~7.5 fold in kinase activity, while the catalytic domain showed a ~35-fold increase compared to FL WT. This result indicates that E225A mutation on the PDZ domain is not sufficient to relieve the autoinhibition of LIMK. Why did the results of in vitro kinase assay and yeast growth assay not match well?

>RESPONSE: We concur that there are no obvious differences in the levels of cofilin pSer between yeast expressing various forms of LIMK that differ in their impact on cell growth and in vitro kinase activity. Growth suppression likely requires that cofilin be phosphorylated to high stoichiometry, or put another way, a relatively low level of residual non-phosphorylated cofilin can likely support growth. In this case, small differences in the level of cofilin pSer3 that could strongly influence the growth rate could be difficult to reproducibly observe by immunoblot. To address this issue, we used an alternative approach using Phos-tag gels, which can resolve phosphorylated from non-phosphorylated cofilin. In this way, we can make a more accurate ratiometric measurement of the fractional phosphorylation of cofilin. Over a time course we found partial phosphorylation at 2 hours post LIMK induction, making it an ideal timepoint to assess differences among the various mutants. We can now clearly see significant differences in the level of cofilin phosphorylation that correlate with the level of activity seen in vitro and the degree of growth suppression (now included as new panels **Fig 4c and d**). We will point out that there is no expectation that there be a linear relationship between activity in vitro and cofilin phosphorylation in cells, where levels of phosphorylation are high and reflect a balance of kinase and phosphatase activities. We have added new discussion of in the text on page 7 as shown below.

Page 7. "To assess whether these alterations in yeast growth were indeed due to changes in LIMK catalytic activity, we **examined the level of cofilin phosphorylation following LIMK induction by Phos-tag SDS-PAGE. We observed that PDZ domain mutations increased the proportion of phosphorylated cofilin in yeast (Figure 4c,d). As this analysis suggested increased kinase activity, we directly assessed the impact of LIMK mutations on phosphorylation of cofilin in vitro.** We purified FL WT LIMK1 and the"

Figure 4: PDZ domain mutants suppress yeast growth and increase cofilin phosphorylation. a) Serial dilutions of *cof1Δ* yeast expressing human cofilin and the indicated human LIMK1 mutants. Controls of human LIMK1 constructs, full-length (FL), kinase domain (CAT), unphosphorylatable cofilin S3A (S3A) and empty vector (EV). Mutants of full-length LIMK1: E225A, D221A, R222A, L165A, Q251A. Corresponding LIMK2 residue shown in parentheses. Five-fold dilutions of yeast cultures were plated on solid media in the presence of glucose (-Gal) or galactose and raffinose (+Gal/Raff) to induce LIMK1 expression. Plates were grown at 30°C

for 2 days (glucose plate) or 4 days (galactose plate). Representative of 3 independent experiments. **b)** Mutants assessed are shown on the cartoon and surface representations of the conservation map of LIMK2 PDZ domain. Residues shown and equivalent human LIMK1 residue numbers: L152 (L165 in LIMK1), Q232 (Q251 in LIMK1), D202 (D221 in LIMK1), R203 (R222 in LIMK1) and E206 (E225 in LIMK1). **c)** Immunoblots of lysates corresponding to yeast plated in (a). Cofilin species are separated based on phosphorylation state by Phos-tag SDS-PAGE. Kss1 serves as a loading control. Images are representative of N=4. **d)** Quantification of immunoblots measuring the percentage of total unphosphorylated cofilin.

Minor points:

) The full name of LIM?

>RESPONSE: Thank you for noticing this accidental omission. We have updated the text as below:

Page 3. "Cytoskeletal remodeling occurs in response to external stimuli and is required for essential processes such as cell invasion, proliferation, cytokinesis, adhesion, and differentiation⁵⁰⁻⁵². Actin severing is necessary for a dynamic cytoskeleton and is regulated by the LIM (**Lin11, Isl-1 & Mec-3) domain kinases (LIMK), which are key effectors of Rho GTPase pathways⁵³⁻⁵⁶**

) In line 195, it should be Figure S5, not Figure 4A.

>RESPONSE: Thank you. We have updated the text to address this typo.

) In line 213, 'most mutants showed an approximate 4-fold increase in kinase activity, with E225A demonstrating a ~7.5-fold increase.' As shown in Figure 5, mutants D221A and R222A showed comparable changes to E225A.

>RESPONSE: We thank the reviewer for pointing out this poorly worded sentence. We have edited the text to make our main point here, that E225A has the highest kinase activity. As per below:

Page 8. "To assess whether these alterations in yeast growth were indeed due to changes in LIMK catalytic activity, we conducted in vitro kinase activity assays. We purified FL WT LIMK1 and the panel of PDZ domain mutants alongside a catalytic domain control from yeast. We assessed phosphorylation of purified human cofilin by these LIMK preparations and found that K175D, a non-conserved mutant, showed no difference in cofilin phosphorylation compared to the WT FL LIMK1. **In contrast, most mutants showed a significant increase in kinase activity, with E225A having the highest increase (Figure 5).** Solubility analysis for the...."

} **The authors examined the phosphorylation of the activation loop of human LIMK WT and PDZ mutants using yeast lysates (Figure 6A) and purified protein from yeast (Figure S7). However, the results of D221A and R222A in Figure 6A and Figure S7 are inconsistent.**

>RESPONSE: We thank the reviewer for allowing us to clarify this seeming divergence in the results presented in the original **Figures 6a and S7 (now S9)**. We believe the difference in observed phosphorylation between the gel presented in the original **Figure S7** and the quantification presented in the original **Figure 6a** to be due to differences in loading (as observed in the FLAG-LIMK1 blot). When we normalize to FLAG signal we find consistent LIMK phosphorylation across experiments. In **Reviewer Figure R2** (shown above) we show the repeats and data that we used to generate **Figure 6a**.

} **In Figure 6A, the top panel used yeast lysates to show the phosphorylation of LIMK1 while the second panel used purified protein to show the protein expression according to the figure legend. And, Q232A is LIMK2 mutant, equivalent to LIMK1 Q251A mutant.**

>RESPONSE: We thank the reviewer for highlighting this typographical error. We have edited the text of the figure legend as below.

Figure 6: Assessment of LIMK activation loop phosphorylation in yeast. **a)** Western Blot assessing activation loop phosphorylation in LIMK1 constructs from our yeast growth assays. The top panel shows yeast lysates blotted with anti-pLIMK antibody and the bottom panel purified yeast protein blotted with anti-FLAG. CAT indicates catalytic domain, FL indicates full-length LIMK1. Mutations of full-length LIMK1 are indicated. **b)** Quantification of Western blot signal. For each construct, phospho-LIMK1 signal was normalized to FLAG signal then compared to full-length LIMK1 (FL). **c)** Quantification of FLAG signal in all four replicates, normalized to FLAG-LIMK1 FL signal. All mutant constructs signals are non-significantly different from each other. Statistical analysis was carried out using a non-parametric unpaired Mann-Whitney test. One star (*) indicates $p < 0.05$. A total of 4 replicates were analyzed using GraphPad Prism.

} **Figure S6, the figure and legend are inconsistent.**

>RESPONSE: This was a typo. We have edited the text of the legend as per below.

Legend for Figure S6. Bacterial expression and solubility tests for LIMK2 PDZ domain mutants. *E. coli* lysate fractionation of the crystallized his-tagged LIMK2 PDZ domain construct, and comparison with PDZ mutants in this construct. Residue number for mutations corresponding for **LIMK2** (top) and equivalent **LIMK1** residue (bottom) are shown.

} **Line 249, ‘reveals an unusual addition to this common fold’?**

>RESPONSE: We thank the reviewer for highlighting this poorly worded phrase. We have updated the prose to make the passage clearer.

Page 9. “Our crystal structure reveals **new structural insights into the well-studied PDZ fold.** Comparison of the LIMK2 PDZ domain...”

REVIEWERS' COMMENTS

Reviewer #1 (Remarks to the Author):

All issues resolved - thank you!

Reviewer #2 (Remarks to the Author):

The interaction between the PDZ domain and the catalytic kinase domain is very weak according to the revision, which is not convincing to draw the conclusion mentioned in line 130-131. Please revise the conclusion.

Response to Reviewers

Reviewer #1 (Remarks to the Author):

All issues resolved – thank you!

>RESPONSE: Thank you for your very helpful comments and queries, we think the paper is much improved due to these.

Reviewer #2 (Remarks to the Author):

The interaction between the PDZ domain and the catalytic kinase domain is very weak according to the revision, which is not convincing to draw the conclusion mentioned in line 130-131. Please revise the conclusion.

>RESPONSE: We appreciate the careful and thoughtful comments and queries and thank the reviewer for this specific comment. As the specific line numbers mentioned do not refer to this comment, we looked to line 231-232 which did contain a comment about intramolecular interactions, and which we believe the reviewer was referring to. We have revised the sentence to the following:

“Overall, we infer that the conserved β A- β F- β D surface of the PDZ domains of LIM domain kinases represents a surface that can impact LIMK kinase activity.”